# Microtubule minus-end aster organization is driven by processive HSET-tubulin clusters

Stephen R. Norris[1,4], Seungyeon Jung[1], Prashant Singh[1], Claire E. Strothman[1], Amanda L. Erwin[1,2,5], Melanie D. Ohi[1,2,5], Marija Zanic[1,3] & Ryoma Ohi[1,2,5]

Higher-order structures of the microtubule (MT) cytoskeleton are comprised of two architectures: bundles and asters. Although both architectures are critical for cellular function, the molecular pathways that drive aster formation are poorly understood. Here, we study aster formation by human minus-end-directed kinesin-14 (HSET/*KIFC1*). We show that HSET is incapable of forming asters from preformed, nongrowing MTs, but rapidly forms MT asters in the presence of soluble (non-MT) tubulin. HSET binds soluble (non-MT) tubulin via its N-terminal tail domain to form heterogeneous HSET-tubulin clusters containing multiple motors. Cluster formation induces motor processivity and rescues the formation of asters from nongrowing MTs. We then show that excess soluble (non-MT) tubulin stimulates aster formation in HeLa cells overexpressing HSET during mitosis. We propose a model where HSET can toggle between MT bundle and aster formation in a manner governed by the availability of soluble (non-MT) tubulin.

[1] Department of Cell and Developmental Biology, Vanderbilt University, Nashville, TN 37232, USA. [2] Department of Cell and Developmental Biology, University of Michigan Medical School, Ann Arbor, MI 48109-2216, USA. [3] Department of Chemical and Biomolecular Engineering, Vanderbilt University, Nashville, TN 37232, USA. [4] Present address: Division of Hematology/Oncology, Vanderbilt University Medical Center, Nashville, TN 37232, USA. [5] Present address: The Life Sciences Institute, University of Michigan Medical School, Ann Arbor, MI 48109-2216, USA. Correspondence and requests for materials should be addressed to M.Z. (email: marija.zanic@vanderbilt.edu) or to R.O. (email: oryoma@umich.edu)

An organized microtubule (MT) cytoskeleton is essential for a broad spectrum of biological processes, ranging from cell polarization to division. A striking example is the mitotic spindle, which drives chromosome segregation[1–3]. MT organization depends on both the intrinsic dynamics of MTs and on cellular factors that govern MT positioning and assembly. MTs are polymers of αβ-tubulin that grow and shrink with a fast-growing (plus) and slow-growing (minus) end[4]. During all phases of the cell cycle, it is estimated that between one-half and one-third of total tubulin (termed soluble tubulin) is not incorporated into MT polymers[5], and instead exists as either αβ-heterodimers or small structural intermediates that remain poorly understood[6]. In the absence of MT-associated proteins (MAPs), MTs are unable to organize into higher-order structures. Collectively, the molecular properties of MAPs on a ~nanometer scale determine the architecture of MT arrays on a ~micron scale through diverse mechanisms including MT cross-linking, translocation, and regulation of MT dynamics.

With the exception of templated structures (e.g., the axoneme), MT structures are built from two basic architectures[7]: (i) MT bundles, which can exist in either antiparallel or parallel configurations, and (ii) MT asters, which are radial arrays of MTs focused at a pole. Both architectures are represented in the mitotic spindle. Interpolar MT bundles facilitate sliding that separates two spindle poles, which are defined by two MT asters. Although bundles minimally require MAP-induced cross-linking for their formation, aster formation requires MT end focusing or a specialized structure to impart radial symmetry[8,9]. In cells, aster formation is driven predominantly by centrosomes[10]. When centrosomes are either present in excess[11] or absent[12,13], cells use other factors to organize MT minus ends within the spindle to ensure its bipolar geometry.

Our knowledge of aster formation mechanisms derives primarily from in vitro work. In mitotic cell extracts, the formation of MT asters requires cytoplasmic dynein, along with its cofactor NuMA[14–17]. Aster formation depends on the ability of dynein to cross-link two MTs[18] and move processively toward MT minus ends, leading to a model where cargo MTs are transported along track filaments[14]. The minus-end-directed kinesin-14 XCTK2 is also capable of driving aster formation from growing MTs, but unlike dynein, XCTK2 requires no additional cofactors[19]. On a molecular level, the mechanisms underlying aster formation by both dynein and kinesin-14 are unclear. This is especially true for kinesin-14s, as they are nonprocessive motors, i.e., they do not move unidirectionally as single motors on a MT track[20–27]. Furthermore, in studies using nongrowing MTs, kinesin-14s form, slide, and sort MT bundles, rather than asters[24,28,29]. It thus remains to be seen how kinesin-14s can simultaneously act as promoters of both bundles and asters, and whether this behavior depends on the state of MTs (i.e., growing vs. nongrowing).

Here, we establish the mechanism of aster formation by HSET/*KIFC1* (human kinesin-14). We determine that, consistent with previous studies, HSET is a nonprocessive motor on the single-molecule level that forms MT bundles. We show that HSET's processivity is caused by the formation of multimotor clusters upon binding to soluble (non-MT) tubulin, and that this processivity is essential for promoting aster formation. Finally, we present data suggesting that HSET's aster-forming behavior can be activated by excess soluble (non-MT) tubulin in mitotic HeLa cells. Our work establishes a general principle of aster formation by MT-cross-linking motor proteins, and demonstrates how motor regulation on a molecular level dictates the formation of higher-order MT structures.

## Results

**Aster and bundle formation by HSET is context-dependent.** To determine if HSET, like its *Xenopus* homolog XCTK2, could form MT asters, we expressed and purified four HSET truncations tagged with an N-terminal enhanced green fluorescent protein (EGFP) (Fig. 1a, Supplementary Figure 1a) for use in self-organization assays[19]. We verified that each construct is a dimer by comparing their stepwise photobleaching traces[30] to EGFP-XMCAK, a known dimer[31] (Supplementary Figure 1b, c). When we combined 100 nM full-length EGFP-HSET, 20 μM tubulin, and saturating adenosine triphosphatase/guanosine-5′-triphosphate (ATP/GTP) at 37 °C, we observed MT aster formation, which reached a steady state within ~10 min (Fig. 1b, top, Supplementary Figure 1d, and Supplementary Movie 1), consistent with published studies using Ncd multimers[9] or XCTK2[19]. Deletion of either the N-terminal tail (EGFP-HSETΔTail), or the conserved C-terminal motor domain (EGFP-HSETΔMotor) eliminated aster formation (Fig. 1b, middle). When the motor domain was replaced by another copy of the N-terminal tail domain (EGFP-HSET-DoubleTail, see[19]), we observed bundle formation, but no MT asters (Fig. 1b, bottom). The entire HSET molecule is thus required for aster formation of growing MTs.

We next asked whether HSET was capable of forming asters from preformed, nongrowing MTs stabilized by GMPCPP[32]. When we combined 100 nM full-length EGFP-HSET with preformed MTs (1 μM polymeric tubulin), we observed no aster formation or pole focusing. Rather, consistent with the established role of kinesin-14 as a MT bundling factor[24,28,29,33], we observed MT bundling and sliding, which reached a steady state within ~20 min (Fig. 1c, top, Supplementary Figure 1e, and Supplementary Movie 2). As expected, deletion of either the tail or motor domain prevented any MT cross-linking (Fig. 1c, middle), while EGFP-HSET-DoubleTail produced long-MT bundles (Fig. 1c, bottom). HSET is thus unable to form asters from preformed MTs in the absence of soluble (non-MT) tubulin.

**Soluble (non-MT) tubulin induces HSET motor processivity.** To test whether HSET is processive on a single-molecule level, we visualized HSET constructs associating with stabilized MT tracks using single-molecule total internal reflection fluorescence (TIRF) microscopy (Fig. 2a). In order to (i) account for the possibility that low ionic strength could weakly activate motor processivity[26], and (ii) increase the short association times of EGFP-HSETΔMotor to an observable level, we performed these experiments in low-salt (P12) buffer (see Methods). Consistent with previous studies on Ncd[24] and a recent study on HSET[29], single molecules of EGFP-HSET and EGFP-HSETΔMotor showed diffusive behavior, whereas deletion of the tail domain (EGFP-HSETΔTail) led to transient, static interactions with the MT (Fig. 2b and Supplementary Movie 3). Mean-squared displacement (MSD) analysis confirmed bidirectional diffusion, with EGFP-HSETΔMotor diffusing more rapidly than EGFP-HSET (Fig. 2c). We concluded that individual HSET dimers diffuse on the MT surface, that the tail domain is critical for rapid diffusion, and that tail deletion is insufficient to form a constitutively active motor. The latter observation suggests that HSET's activity is unlikely to be dictated by an autoinhibition mechanism, where the tail domain would interfere with motor activity to prevent futile ATP hydrolysis[34].

We considered the possibility that HSET's inability to form MT asters from preformed MTs depends on the presence of soluble (non-MT) tubulin. To investigate the effect of tubulin on the motor, we examined EGFP-HSET's motility on stabilized MTs in the

presence of 2 µM tubulin (Fig. 2d). At low (1 nM) HSET concentrations, tubulin addition had little effect (Supplementary Figure 2a). However, with increasing HSET concentrations (10 and 20 nM), the addition of tubulin increased the frequency of processive events (Fig. 2e and Supplementary Movie 4). Our quantification of processive events with 20 nM EGFP-HSET is likely an underestimate because it was difficult to resolve individual events at this high frequency (Supplementary Figure 2b).

In order to visualize individual motility events in these conditions of high motor concentration, we performed a spike-in TIRF experiment where 5% of HSET particles contained the N-terminal EGFP-tag at a total HSET concentration of 10 nM (Supplementary Figure 2c). MSD analysis confirmed that HSET molecules shifted from the characteristic diffusive behavior to processive, unidirectional motion upon the addition of 2 µM tubulin (Supplementary Figure 2d). These results suggest that HSET is capable of two configurations: one where it diffuses bidirectionally on the MT lattice, and another where it travels unidirectionally in a processive state, and that the addition of tubulin promotes this second state.

**HSET transports soluble (non-MT) tubulin to MT minus ends**. To investigate the fate of tubulin, we mixed 10 nM Cy5-tubulin with 10 nM unlabeled HSET and visualized the Cy5 channel using TIRF (Supplementary Figure 2e). Under these conditions, we observed Cy5-tubulin to move processively with a velocity of $8.9 \pm 0.2$ µm/min (mean $\pm$ 95% CI, $n = 362$), and run length of $4.2 \pm 0.6$ µm (mean $\pm$ 95% CI, $n = 150$) (Fig. 2f). These processive particles dwelled at MT minus ends with a mean dwell time of $25 \pm 8$ s (mean $\pm$ 95% CI, $n = 50$) (Supplementary Figure 2f). This is consistent with the published velocities and run lengths of teams of Ncd motors tethered by a double-stranded DNA scaffold[27], but this movement is far more processive than individual Ncd motors[26,27]. Near-simultaneous imaging of 1 nM EGFP-HSET and 100 nM Cy5-Tubulin by two-color TIRF showed that EGFP-HSET moved unidirectionally only when colocalized with Cy5-tubulin (Fig. 2g and Supplementary Movie 5). Conversely, we never observed Cy5-tubulin transport by EGFP-HSETΔTail, suggesting that the tail domain of HSET plays a critical role in tubulin transport (Supplementary Figure 3a).

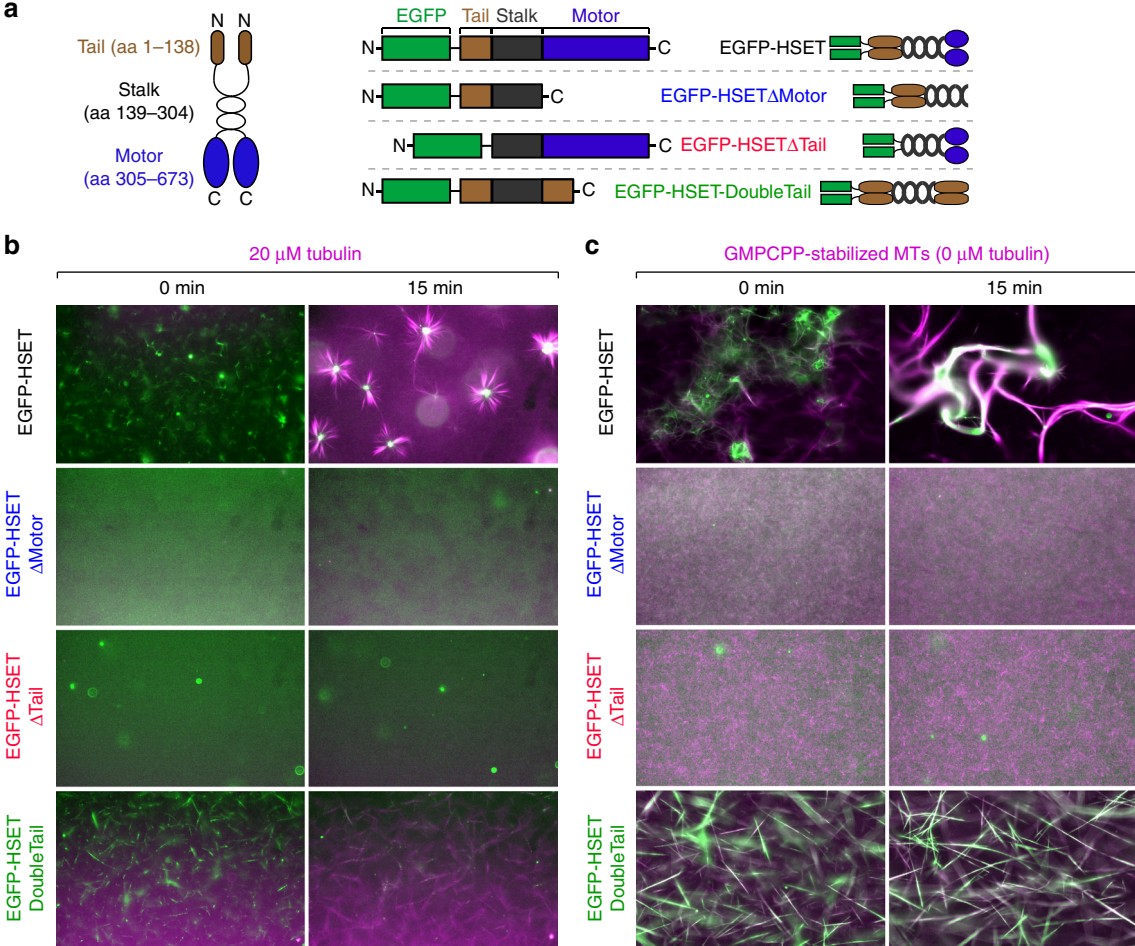

**Fig. 1** Full-length HSET organizes growing MTs into asters. **a** Schematic of HSET truncations purified in this study. HSET contains two MT-binding domains: an ATP-independent globular tail domain located at the N terminus (amino acid 1–138, brown), and an ATP-dependent conserved kinesin motor domain located at the C terminus (aa 305–673, blue). HSET also contains a coiled-coil stalk domain necessary for dimerization (aa 139–304, black). All constructs contained an N-terminal 6× His tag used for affinity purification. **b** Aster formation of growing MTs by HSET. 20 µM tubulin (10% Alexa594-labeled, magenta) was mixed in assay buffer with the indicated EGFP-HSET truncation (green) and monitored by time-lapse microscopy at 37 °C. With the exception of EGFP-HSETΔTail (20 nM), all HSET constructs were present at 100 nM. **c** Bundle formation of nongrowing, GMPCPP-stabilized MTs by HSET. Alexa594-labeled GMPCPP-MTs (10% labeled, 1 µM tubulin in polymer form, magenta) were mixed in assay buffer with the indicated EGFP-HSET truncation (green) and monitored by time-lapse microscopy at 37 °C. HSET concentrations are identical to **b**. For contrast measurements over time, see Supplementary Figure 1d, e. For movies, see Supplementary Movies 1–2. For additional EGFP-HSET images on GMPCPP-MTs, see Fig. 4a. Technical replicates of experiments in **b**, **c** were repeated $n \geq 3$ times for each condition, and representative images are shown. Scale bars, 50 µm

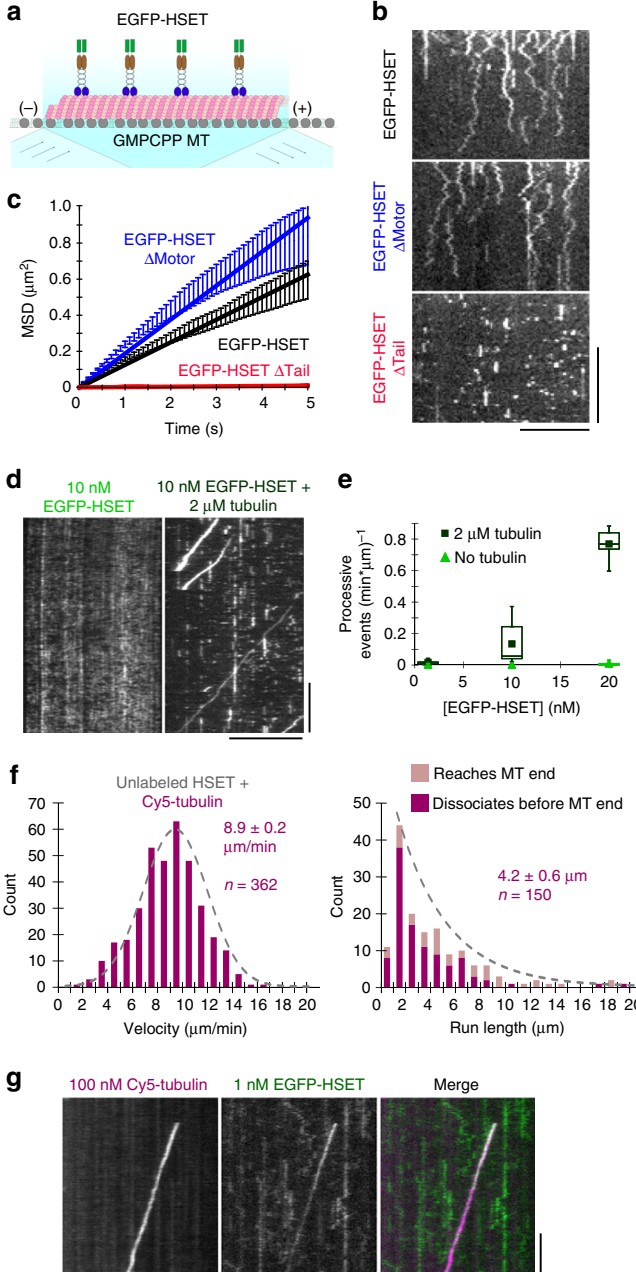

**Fig. 2** Soluble (non-MT) tubulin activates processive motility of HSET on single MTs. **a** Schematic. EGFP-HSET truncations were diluted in P12 buffer and monitored on GMPCPP-stabilized MTs by time-lapse TIRF. **b** Representative kymographs for time-lapse TIRF images for the indicated constructs at the following concentrations: EGFP-HSET and EGFP-HSETΔMotor, 50 pM. EGFP-HSETΔTail, 250 pM. Distance is on the x-axis (scale bar, 10 μm), and time is on the y-axis (scale bar, 10 s). **c** Mean-squared displacement (MSD) analysis of particle motion. The reported diffusion constant $D$ is determined from a linear fit over the first 5 s, with the units $nm^2/s$: EGFP-HSET: $D = 6.3 \times 10^4$, $n = 206$; EGFP-HSETΔMotor: $D = 9.4 \times 10^4$, $n = 197$; EGFP-HSETΔTail: $D = 0.1 \times 10^4$, $n = 200$. Data are presented as the calculated mean MSD (y-axis) from two independent experiments over the indicated time intervals (x-axis) for the indicated $n$ particles ± SEM. **d** EGFP-HSET in BRB80 + 50 mM KCl was observed in the absence (left) or presence (right) of soluble tubulin and visualized by kymograph (x-scale bar, distance, 10 μm; y-scale bar, time, 1 min). **e** Quantification of processive (≥5 s) event frequency as a function of [EGFP-HSET] in the presence (dark green) or absence (light green) of 2 μM tubulin. Data are presented as the number of processive events divided by the total observed MT length multiplied by the observation time for two independent experiments ± SD calculated from $N \geq 10$ movies for each condition. Boxes represent first and third quartiles, whiskers represent detection limits, and lines represent median (mean overlaid). **f** Unlabeled HSET was mixed with 10 nM Cy5-tubulin in BRB80 + 50 mM KCl and observed. Velocities and run lengths of moving Cy5-tubulin particles were determined by kymograph and plotted as histograms. Data are reported as the mean velocity and run length values of $n$ particles from CDF fitting ± the 95% CI from bootstrapping from two independent experiments. **g** 100 nM Cy5-tubulin (magenta) and 1 nM EGFP-HSET (green) were observed near-simultaneously by high-speed TIRF in BRB80 + 50 mM KCl, and visualized by kymograph (x-scale bar, distance, 5 μm; y-scale bar, time, 10 s)

moving particles appeared as diffraction-limited spots under our imaging conditions (Supplementary Figure 3b). The fluorescence intensities of moving EGFP-HSET particles displayed a broad distribution, which was similar whether tubulin was present at a stoichiometric excess of 20,000:1 (Fig. 3b, top) or 200:1 (Fig. 3b, bottom). These bright, moving particles are unlikely to be aggregates of EGFP-HSET because: (i) EGFP-HSET particles with intensity >80,000 a.u. were absent from thousands of counted particles for EGFP-HSET alone, and (ii) EGFP-HSET photo-bleaching events containing >2 steps were very infrequent ($n = 7/206$ particles, see Supplementary Figure 1c).

We used an analogous approach to assess the number of transported tubulin molecules (Fig. 3c). Similarly to EGFP-HSET, the first frame of moving Cy5-tubulin particles displayed significantly (11.8-fold) brighter intensity on average than single Cy5-tubulin molecules alone. These bright particles were absent from thousands of counted individual Cy5-tubulin molecules, were similarly diffraction-limited (Supplementary Figure 3c), and their fluorescence intensity distribution did not depend on guanine nucleotide state of tubulin (Fig. 3d). To ensure that aggregates of tubulin were absent from our preparation, we performed analytical ultracentrifugation (AUC) on our purified tubulin (Supplementary Figure 3d). We did not observe any significant population with a higher sedimentation coefficient than our dimer fraction, and thus concluded that our tubulin was free of aggregates.

Collectively, these data show that HSET-tubulin clusters are heterogeneous, containing multiple motors and multiple tubulin molecules. To further investigate the nature of HSET-tubulin complexes, we mixed 5 μM tubulin with 200 nM EGFP-HSET and subjected this mixture to AUC (Supplementary Figure 3e). Importantly, we did not observe a significant sedimentation signal beyond the prominent dimer peak in these experiments, suggesting that the population of HSET-tubulin clusters in

To test whether the tail domain directly interacts with soluble (non-MT) tubulin, we performed a co-immunoprecipitation of purified tubulin and HSET truncations via the common EGFP-tag (Fig. 3a, Supplementary Figure 7). We detected higher than background levels of tubulin in reactions containing EGFP-HSET, EGFP-HSETΔMotor, and EGFP-HSET-DoubleTail, whereas minimal tubulin was detected for EGFP-HSETΔTail or when no HSET was present. These results demonstrate that soluble (non-MT) tubulin binds the HSET tail domain directly.

**Tubulin-HSET clusters are highly processive**. Processive movement of kinesin-14 motors was previously reported for multimers of Ncd motors[26,27]; we thus wondered if the processive tubulin-HSET particles contain multiple HSET motors. We used a fluorescence intensity-based approach to quantify the number of motors in a moving particle (Fig. 3b). Moving EGFP-HSET particles were noticeably brighter (~3–4 × on average) than single EGFP-HSET particles adhered to a glass cover surface, and these

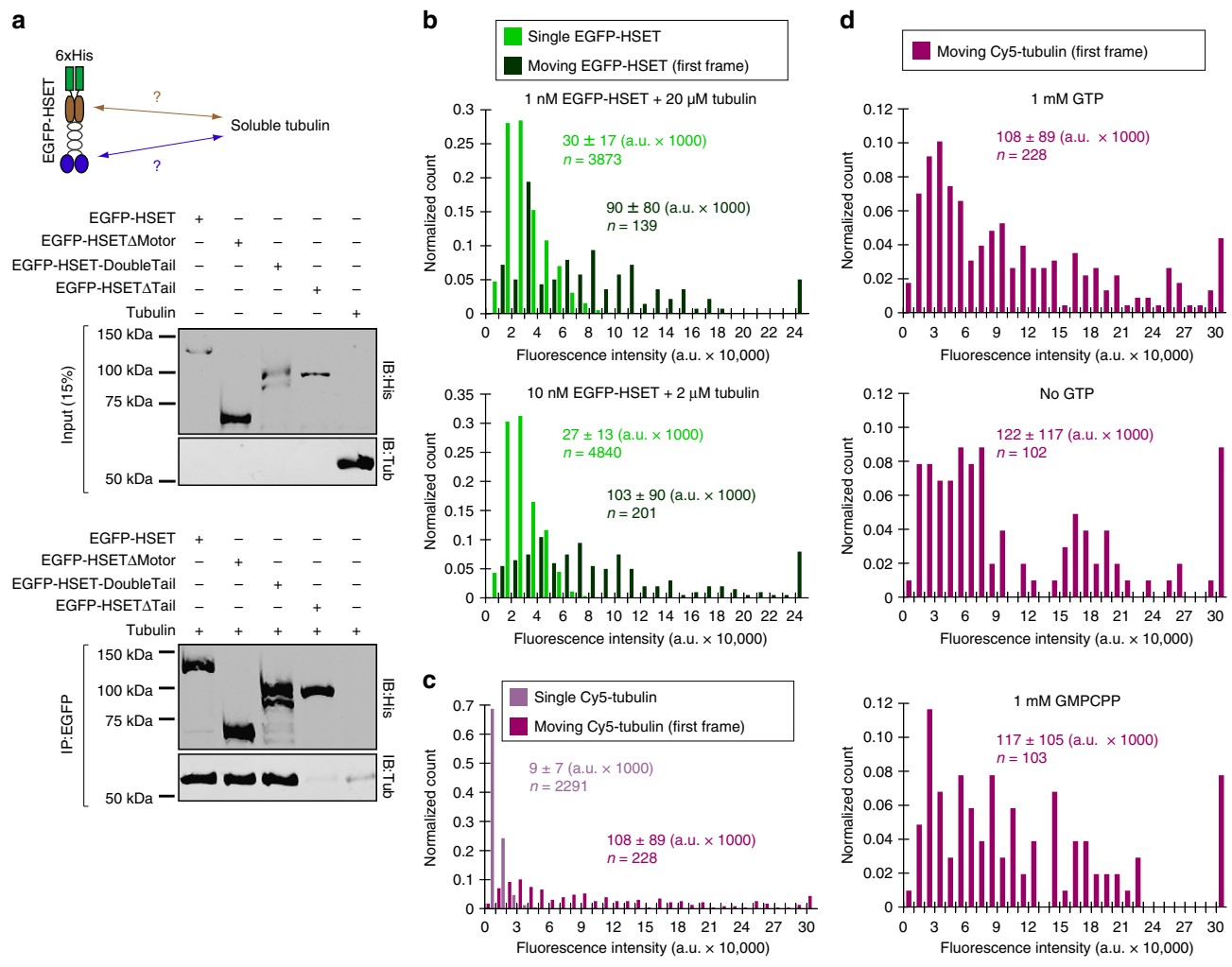

**Fig. 3** Soluble (non-MT) tubulin binding to N-terminal HSET tail domain induces HSET-tubulin clustering to activate long-range unidirectional motion. **a** Co-immunoprecipitation of purified HSET with soluble tubulin. The indicated EGFP-HSET truncation (50 nM concentration) was incubated with soluble tubulin (250 nM concentration) and the common EGFP-tag was used for immunoprecipitation. Inputs were loaded at 15% of total protein. A representative blot from $N = 2$ independent experiments is shown. **b** Fluorescence intensity analysis of EGFP-HSET diluted to single-molecule levels and adhered to a glass cover slip (light green) compared to the first frame of moving EGFP-HSET after the addition of tubulin (dark green, concentrations indicated). The fluorescence intensity of individual EGFP-HSET particles was determined by Gaussian fit, and intensity distributions were plotted as histograms for the population, where normalized count is the observed value in the bin divided by the total number of particles. Data are reported as the arithmetic mean ± SD for the indicated $n$ particles from 2 independent experiments, where $N \geq 4$ movies for each condition. **c, d** Fluorescence intensity analysis of Cy5-tubulin diluted to single-molecule levels and adhered to a glass cover slip (light purple) compared to the first frame of moving Cy5-tubulin (10 nM) in the presence of 10 nM unlabeled HSET (dark purple). The fluorescence intensity of individual Cy5-tubulin particles was determined by Gaussian fit, and intensity distributions were plotted as histograms for the population. Data are reported as the arithmetic mean ± SD for the indicated $n$ particles from 2 independent experiments where $N = 6$ movies for moving Cy5-tubulin. For **d**, fluorescence intensities were determined in the presence of the indicated guanine nucleotide, where $N \geq 2$ movies for each condition. All experiments were performed in BRB80 + 50 mM KCl with 0.5 mg/mL casein

solution is relatively small. Alternatively, the absence of resolvable high-molecular weight peaks could be a result of the heterogeneous composition of HSET-tubulin clusters. Regardless, this small population of clusters appears to be highly enriched on MTs, as observed in our TIRF experiments (Supplementary Figure 2b).

**HSET-tubulin clusters drive self-organization of MT asters.** Because soluble (non-MT) tubulin promotes HSET's processivity by inducing cluster formation, we tested whether soluble tubulin could promote the ability of HSET to drive the organization of asters using preformed MTs. Upon the addition of 20 μM tubulin to preformed MTs and HSET, we observed rapid formation of radially symmetric asters (~8 min) (Fig. 4a, bottom,

Supplementary Figure 4a, and Supplementary Movie 6). In contrast, with the addition of 2 μM tubulin, we observed structures of varying morphologies, having characteristics of both bundles and asters (Fig. 4a, middle).

To investigate the potential role of MT polymerization in aster formation, we performed self-organization experiments under conditions incompatible with MT assembly. Specifically, we: (i) introduced saturating (100 μM) levels of colchicine, a tubulin-sequestering drug that prevents polymerization, (ii) replaced 1.5 mM GTP with 1.5 mM GDP, a guanine nucleotide that is incompatible with polymerization, and (iii) omitted taxol, a field-standard component for self-organization buffer[19]. Here, we again observed rapid formation (~5 min) of MT asters in the presence of 20 μM tubulin (Fig. 4b, Supplementary Figure 4b). We obtained similar results with saturating (33 μM) amounts of

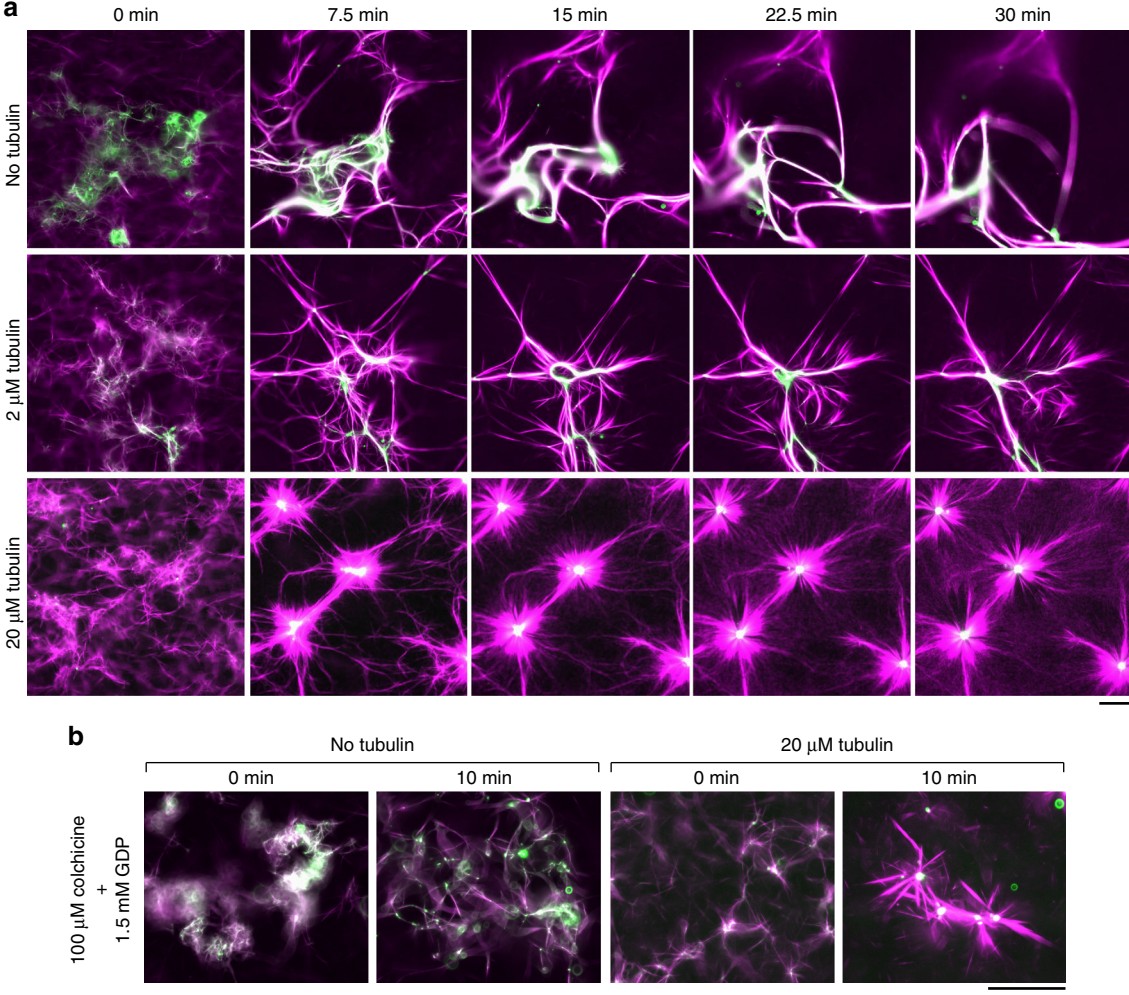

**Fig. 4** Soluble (non-MT) tubulin promotes the ability of HSET to drive aster self-organization of GMPCPP-MTs independent of MT polymerization. **a** EGFP-HSET-driven self-organization of GMPCPP-stabilized MTs with increasing tubulin concentration. Alexa594-labeled GMPCPP-MTs (10% labeled, 1 μM tubulin in polymer form, magenta) were mixed in assay buffer with EGFP-HSET (100 nM, green) and monitored by time-lapse microscopy at 37 °C. Unlabeled tubulin was added to the reaction at the indicated concentration. Technical replicates were repeated $N \geq 3$ times for each condition, and representative images are shown. Scale bar, 50 μm. **b** EGFP-HSET-driven self-organization of GMPCPP-stabilized MTs in the absence of MT polymerization. Experiments were performed identically to **a** but in the absence of taxol and the presence of saturating colchicine and GDP to prevent polymerization. Technical replicates were repeated $N \geq 2$ times for each condition, and representative images are shown. Scale bar, 50 μm

nocodazole, an alternative tubulin-sequestering drug (Supplementary Figure 4c). These results show that MT polymerization is not required for MT aster formation by HSET.

**HSET clustering (rather than tubulin) drives aster assembly.** We next wanted to determine if multimotor clusters of HSET could lead to aster formation of preformed MTs in the absence of soluble (non-MT) tubulin. To construct HSET clusters without tubulin-induced activation, we incubated 6× His-tagged EGFP-HSET or EGFP-HSETΔTail with streptavidin-Quantum Dots (QDots, ~25 nm diameter) at a 3:1 ratio in the presence of biotin-anti-His antibody (Fig. 5a). After using fluorescence intensity analysis to confirm that ~2–3 motors were present on QDots (2.3 for EGFP-HSET and 3.4 for EGFP-HSETΔTail, on average) (Supplementary Figure 5a), we investigated their motility by TIRF (Fig. 5a and Supplementary Movie 7). As expected, both EGFP-HSET-QDots and EGFP-HSETΔTail-QDots moved processively, with run lengths of $3.7 \pm 0.6$ μm (mean ± 95% CI, $n = 185$) and $4.0 \pm 0.6$ (mean ± 95% CI, $n = 197$) μm, respectively (Fig. 5b,c). Both constructs were capable of dwelling at minus ends of MTs before unbinding. Notably, however, EGFP-HSET-QDots

displayed a ~5-fold enhanced end-dwelling capacity over the EGFP-HSETΔTail-Qdots ($37 \pm 16$ s vs. $7 \pm 2$ s) (mean ± 95% CI, $n = 113$ and 133, respectively) (Fig. 5d).

When we introduced EGFP-HSET-QDots to preformed MTs, asters formed over ~20 min (Fig. 5e, top and Supplementary Movie 8). We directly observed transport of MTs (or MT bundles) along other MTs by EGFP-HSET-QDots during the process of aster formation (Fig. 5f). Interestingly, EGFP-HSETΔ-Tail-QDots were unable to form asters (Fig. 5e, bottom). Because the tail domain was required for robust end-dwelling on MTs (Fig. 5d), this is consistent with a model in which motors must remain end-localized in order to organize MT asters[9]. Alternatively, HSET's tail may be required to bind polymer MTs as a cargo during translocation to minus ends (see Discussion). Regardless, we conclude that cluster formation of EGFP-HSET, whether by addition of tubulin or binding to QDots, leads to aster formation from preformed MTs.

**Non-MT tubulin promotes aster formation by HSET in cells.** Based on our in vitro findings, we hypothesized that the activity of HSET could be modulated by the concentration of soluble

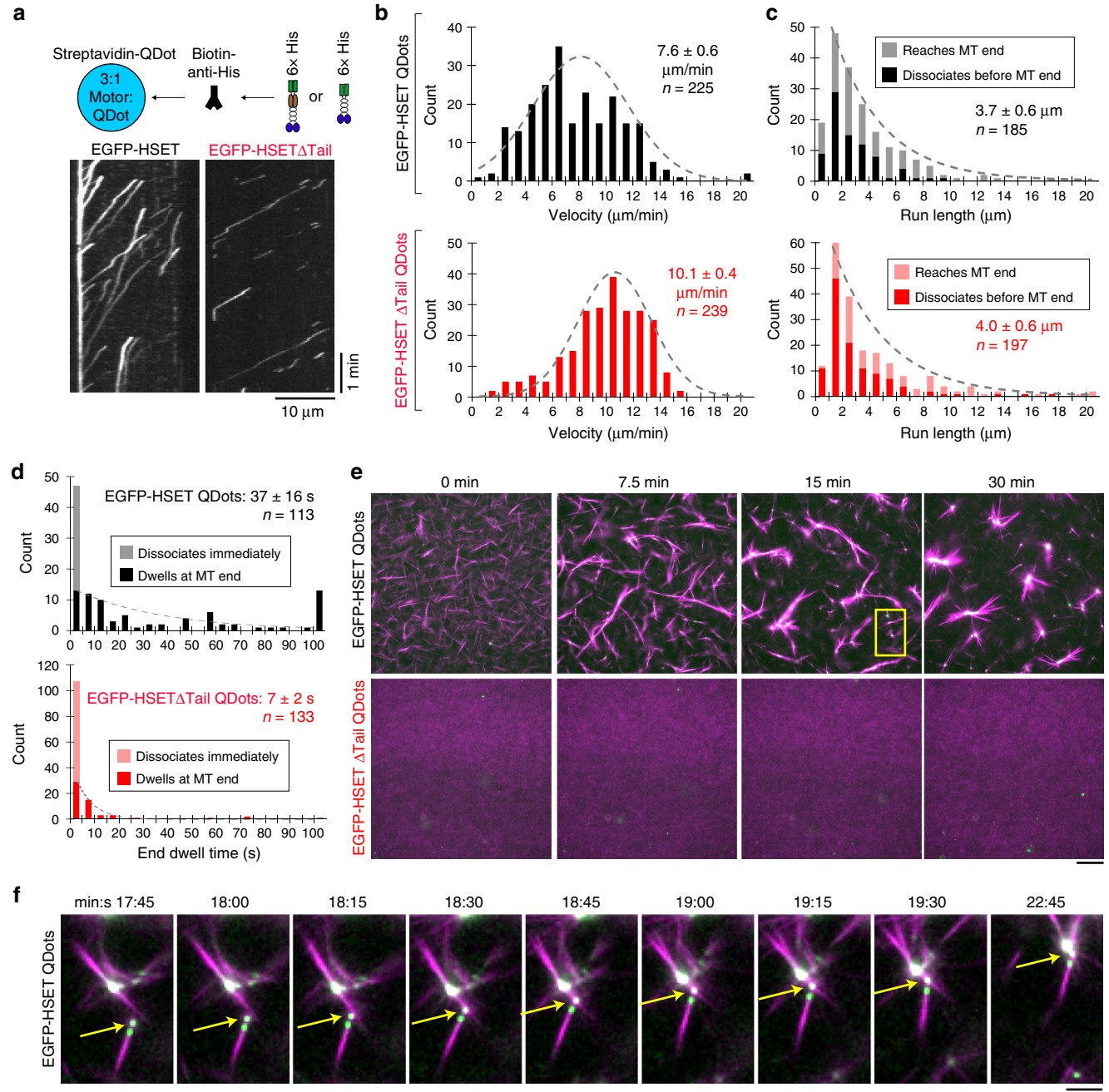

**Fig. 5** Multiple HSET motors conjugated to quantum dots drive self-assembly of GMPCPP-MTs into asters. **a** EGFP-HSET or EGFP-HSETΔTail was conjugated to streptavidin-QDots via the N-terminal 6× His-tag and a biotin anti-His antibody at a 3:1 ratio and visualized via TIRF. Representative kymographs of EGFP-HSET-QDots (left, 1 nM EGFP-HSET: 0.33 nM QDot) and EGFP-HSETΔTail-QDots (right, 0.5 nM EGFP-HSETΔTail: 0.17 nM QDot) are shown (x-scale, distance, 5 µm; y-scale, time, 10 s). **b–d** Velocities (**b**), run lengths (**c**), and end dwell times (**d**) for the indicated constructed conjugated to QDots at a 3:1 ratio (EGFP-HSET, black, EGFP-HSETΔTail, red) were determined by kymograph analysis and plotted as histograms for the population. Data are reported as the mean values (insets) from CDF fitting ± the 95% CI from bootstrapping for the indicated n particles from 2 independent experiments, where N ≥ 4 movies for each condition. Populations for EGFP-HSET-QDots (black, upper) and EGFP-HSETΔTail-QDots (red, lower) are shown. For run length/end dwell times, particles reaching the end of MTs/dissociating immediately (<1 frame) are color-coded on the histograms. **e** Self-organization of GMPCPP-stabilized MTs by EGFP-HSET-QDots and EGFP-HSETΔTail-QDots. Alexa594-labeled GMPCPP-MTs (10% labeled, 1 µM tubulin in polymeric form, magenta) were mixed in assay buffer with the indicated motor-QDot complexes (21:7 nM motor :QDots, green) and monitored by time-lapse microscopy at 37 °C. The yellow box indicates the field of view depicted in **f**. Technical replicates were repeated n ≥ 3 times for each condition, and representative images are shown. Scale bar, 50 µm. **f** Zoomed-in view of the indicated field. The yellow arrow indicates EGFP-HSET-QDots that have accumulated on the minus end of an MT bundle. Time is indicated in min:s. Scale bar, 10 µm

(non-MT) tubulin in cells. We investigated this hypothesis by changing the relative concentrations of both HSET and soluble (non-MT) tubulin during mitosis. To increase HSET protein levels, we used recombination-mediated cassette exchange[35] to

generate a HeLa cell line that inducibly overexpresses EGFP-HSET. By immunoblotting, we observed that EGFP-HSET reached a maximum expression level of ~4-fold overexpression relative to endogenous HSET after 3 days' treatment with

doxycycline (compare to ~2.7-fold overexpression in 1 day) (Fig. 6a, Supplementary Figure 7). Consistent with previous overexpression studies[33], EGFP-HSET localized uniformly to spindle MTs and caused a slightly elongated and tapered spindle morphology after overexpression (Fig. 6b).

To increase soluble (non-MT) tubulin concentrations, we used nonsaturating amounts (500 nM) of nocodazole to partially depolymerize MTs. We treated cells either containing endogenous (Fig. 6c, top) or 4-fold overexpressed (Fig. 6c, bottom) levels of HSET with nocodazole (500 nM) for 15 or 30 min, then fixed and stained these cells for tubulin and centrin, a marker of centrosomes. In the HSET-overexpressing cells, we observed the formation of distinct MT asters, which were positive for both tubulin and HSET at both 15 and 30 min timepoints (Fig. 6c).

Supplementary Figure 6b, c). Over 60% of EGFP-HSET expressing cells contained at least one acentrosomal aster after nocodazole treatment as assessed by centrin immunostaining (Fig. 6c, right). This aster formation after nocodazole treatment was specifically enhanced by EGFP-HSET overexpression, as acentrosomal asters were only observed in <10% of control cells treated with vehicle (Fig. 6d). The observation that nocodazole promotes a basal level of aster formation is consistent with published results[36], as endogenous aster-promoting factors are present in these cells. Finally, we used a live-imaging approach to observe EGFP-HSET during aster formation over time. Following the addition of nocodazole to cells overexpressing EGFP-HSET, we observed a conversion of bipolar spindles to multiple individual asters over 15–30 min (Fig. 6e, Supplementary

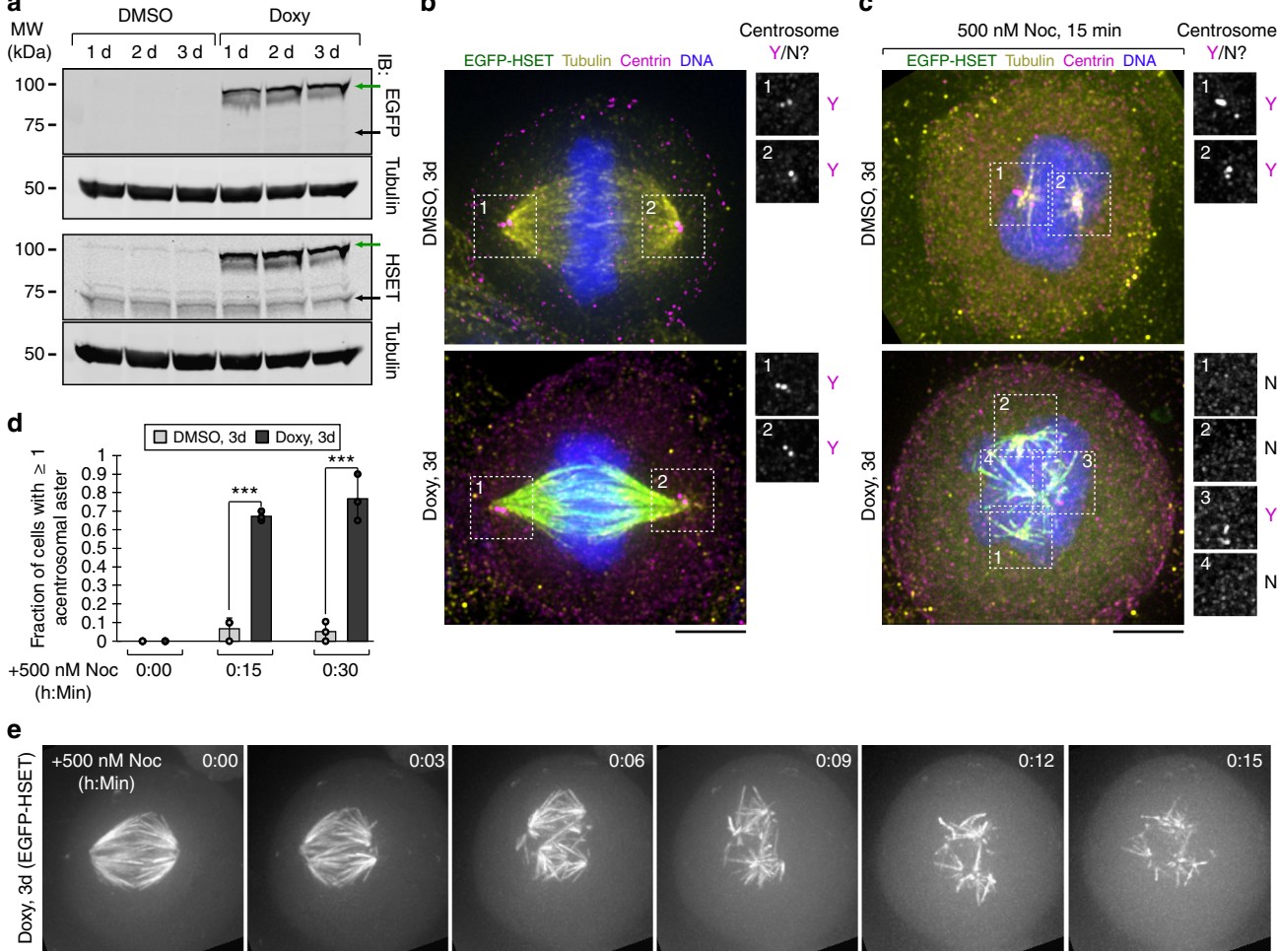

**Fig. 6** Increasing the relative level of soluble (non-MT) tubulin promotes HSET-driven aster formation in cells. **a** Generation of a transgenic HeLa cell line where EGFP-HSET expression is under control of doxycycline. Western blot of whole cell lysates probed with antibodies that recognize EGFP, tubulin, and HSET. Numbers indicate days of induction with doxycycline/DMSO. Green arrow indicates overexpressed EGFP-HSET, and black arrow represents endogenous HSET. **b** Maximum intensity z-projections of metaphase EGFP-HSET HeLa cells treated with DMSO (top) or doxycycline (bottom) for 3 d, then fixed and stained with antibodies against tubulin (yellow), centrin (magenta) or Hoechst (DNA, blue). Numbered boxes correspond to spectrally unmixed centrin-stained regions shown on right. Y/N indicates the presence of centrin at spindle poles. Scale bar, 5 μm. **c** EGFP-HSET HeLa cells were pretreated with DMSO (top) or doxycycline (bottom) for 3 d, then treated with 500 nM nocodazole (Noc) for 15 min to increase the relative levels of soluble tubulin. Cells were fixed, stained, and imaged identically to **b** and maximum intensity z-projections are shown. Scale bar, 5 μm. **d** Quantification of **b**, **c**. After analyzing the tubulin/centrin channels, the fraction of cells containing ≥1 acentrosomal aster was compared between 3 d doxycycline/DMSO cells after treatment with 500 nM nocodazole for 0, 15, and 30 min. Overlaid dots represent the average observed fraction for each independent experiment. ***p < 0.001 by two-tailed t-test. Data were reported as the average ± SEM of N = 3 independent experiments with n ≥ 60 cells for each condition. **e** EGFP-HSET HeLa cells were treated with doxycycline for 3 days to induce maximal expression, and 500 nM nocodazole was added to increase relative levels of soluble tubulin. Maximum intensity z-projections were acquired in the EGFP channel every 3 min and a representative time-course is shown for the indicated times. Scale bar, 5 μm

Movie 9). These experiments show that aster formation in mitotic cells can be modulated by the simultaneous increase of both HSET and soluble (non-MT) tubulin levels.

## Discussion

In this study, we have shown that single HSET motors are minimally processive on MTs, consistent with previous reports on the fly kinesin-14 Ncd[20–27]. This result is also consistent with a recently published study where HSET was shown to diffuse bidirectionally on single MTs[29]. In spite of these unremarkable single-molecule properties, we have shown that kinesin-14s can assemble and remodel the two basic MT architectures employed in higher-order organization: bundles and asters. Our data suggest that aster formation occurs in the presence of soluble (non-MT) tubulin, which causes the clustering of HSET to promote motor processivity. We thus propose that the activity of HSET is context-dependent, being influenced directly by the availability of soluble (non-MT) tubulin vs. MT polymer, and therefore inextricably linked to the dynamic rearrangement of the cytoskeleton during cell division.

The observation that soluble (non-MT) tubulin promotes HSET's processivity suggests a distinct mechanism of kinesin motor activation. Many processive kinesins are regulated by an autoinhibition mechanism, where cargo binding to the tail domain overcomes an inhibitory tail–motor interaction[34]. Three pieces of evidence suggest that HSET is not regulated in this way: (i) tail deletions do not result in activation of either Ncd[20–25] or HSET (this study), (ii) Ncd shows nonprocessive behavior whether bound to single MTs or to MT overlaps where the tail is engaged[24], and (iii) HSET adopts an open conformation when visualized by negative stain EM[37]. Rather, our data demonstrate that HSET's processivity is promoted by motor multimerization upon binding soluble (non-MT) tubulin to form clusters. It is likely that cluster formation occurs in solution, rather than on the MT surface, as we did not observe fluorescence intensity increasing within clusters over time (Fig. 2g). After binding to MTs, both HSET-tubulin and HSET-QDot clusters displayed high processivity. This behavior is consistent with two recent studies reporting that artificial coupling of small numbers of kinesin-14 motors (>2 motors) converts their motility from non-processive to processive[27,38]. This observation has led to speculation that clustering of kinesin-14 motors drives MT-based retrograde transport in plants, which lack cytoplasmic dynein[38,39]. However, while processive kinesin-14 clusters have been observed in plants[38], the physiological clustering agent(s) is unknown. Here, we identify soluble tubulin (where we are defining soluble tubulin as tubulin present in solution under depolymerizing conditions, i.e., not a classic MT polymer) as a physiologically relevant clustering agent for HSET.

What is the population of tubulin that activates HSET's processivity? Because tubulin activates HSET's processivity under conditions that are not compatible with MT polymer formation (i.e., saturating nocodazole, colchicine, and no guanine nucleotide (Fig. 3d, Supplementary Figure 4b, c), MT polymers are unlikely to be the sole activator of motor activity. Instead, our data demonstrate that the remaining pool of soluble (non-MT) tubulin is responsible for modulating motor activity. One outstanding question is whether HSET directly induces the formation of small tubulin oligomers, or whether the HSET tail domain binds a pre-existing population of oligomeric tubulin in solution. In addition to canonical αβ-heterodimers and MT polymers, multiple lines of study argue for the presence of a third population that exists as small oligomers[40,41]. This population is relatively rare and requires ultra-sensitive techniques for its detection (i.e., electron microscopy and fluorescence correlation spectroscopy)[40].

Analogous to conditions where we observed HSET's activation by tubulin (Fig. 3d), these tubulin intermediates form in a GTP-independent manner[40]. That we did not detect a population of tubulin oligomers either by analytical centrifugation or single-molecule fluorescence intensity suggests that this population is exceedingly rare in our experimental conditions. Thus, HSET must either: (i) possess an enhanced affinity for rare tubulin oligomers in solution, or (ii) HSET induces the formation of tubulin oligomers. The order of events in the formation of HSET-tubulin clusters is an important issue that will be addressed in future studies.

How does soluble (non-MT) tubulin promote HSET's aster formation activity? Soluble tubulin has previously been shown to regulate the MT-severing protein katanin by disrupting its ability to bind MT polymer[42]. HSET's ability to toggle between MT bundling and processive transport depends on its N-terminal tail domain, which binds both MT polymer[24,26,28,43,44] and soluble (non-MT) tubulin (this work) (Fig. 7a). In the absence of soluble tubulin, the tail domain binds MT polymer, which results in bidirectional, symmetrical diffusion along a single MT (Fig. 7b, top) and the formation of MT bundles in a many-MT case (Fig. 7b, bottom), as previously demonstrated for Ncd[24,28]. In contrast, when soluble (non-MT) tubulin is present in sufficient quantities (Fig. 7c), the tail domain is now able to bind either soluble (non-MT) tubulin or MT polymer. Binding to soluble (non-MT) tubulin induces HSET-tubulin clusters, which contain multiple motors, break symmetry by moving processively toward MT minus ends, and dwell at these ends. In a many-MT context, HSET sorts these MTs into radial asters, as previously demonstrated for XCTK2[19]. At intermediate tubulin concentrations, we observe structures containing features of both architectures (Fig. 4a). Thus, our model reconciles the observation that kinesin-14 causes locked parallel MT bundles in vitro in the absence of tubulin[24], with its cellular role in organizing MT minus ends at spindle poles[45,46].

What is the molecular nature of HSET-tubulin clusters? Based on our fluorescence intensity measurements, these clusters appear to be heterogeneous in size, composition, and stoichiometry. Our AUC results suggest that either cluster heterogeneity obscures peak detection, or only a small number of clusters is sufficient to drive aster organization. Because these clusters are diffraction-limited, the geometrical nature of these clusters is also yet to be determined. Regardless, we have determined a number of key features for these clusters: (i) they contain multiple HSET motors (possible range of 2–6) and multiple tubulin dimers (possible range of 3–26), (ii) cluster formation does not depend on GTP-driven tubulin assembly, and (iii) clusters are likely not the result of an in vitro aggregation artifact. Importantly, we have shown that HSET's activity in cells is modulated by the intracellular levels of soluble (non-MT) tubulin, providing evidence that our in vitro observations are relevant within the cell.

Our observation that QDots bound to tailless HSET do not form either bundles or asters points to two additional, critical roles for the HSET tail domain in higher-order MT organization. First, it is currently unclear how cargo MTs are engaged by HSET-tubulin clusters to initiate transport (Fig. 7c, bottom). Although cargo MTs may be attached through an available unbound motor domain, it is likely that free HSET tail domains within clusters (i.e., not bound to either soluble (non-MT) tubulin or the lattice) engage cargo MTs directly. Second, our observation that the tail domain enhances MT end-dwelling ~5-fold points to the necessity of cluster end-dwelling in aster formation[9]. Detailed structural investigations of these tail-mediated interactions thus present an exciting direction for future study.

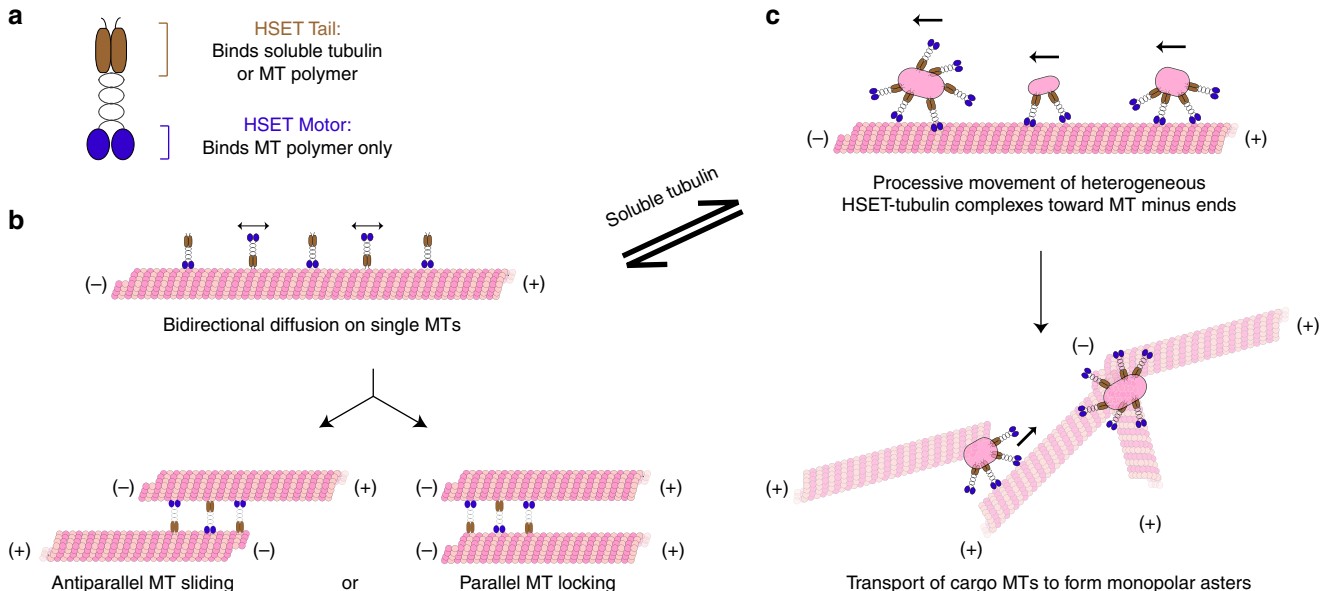

**Fig. 7** Proposed model for MT aster formation by HSET. **a** HSET's N-terminal tail domain is able to bind either soluble (non-MT) tubulin or MT polymer (brown), whereas HSET's C-terminal motor domain is only able to bind MT polymer (blue). The relative availability of soluble tubulin toggles HSET's activity between the following states: **b** HSET is unbound to soluble (non-MT) tubulin and exists as a dimer. This motor is: (i) nonprocessive on single MTs, and (ii) provides MT–MT sliding forces within bundles of MTs similar to established models for Ncd. **c** HSET is bound to soluble (non-MT) tubulin and exists in heterogeneous multimotor HSET-tubulin clusters. The precise molecular and geometrical nature of these clusters remain to be determined. These motors: (i) move processively toward the minus ends of single MTs, and (ii) processively transport MTs or MT bundles along existing MT tracks to form MT asters. This ability to toggle between modes relies on the unique ability of the N-terminal tail domain to bind both soluble (non-MT) tubulin and MT polymer

In summary, our data demonstrate how the interplay between tubulin and HSET can lead to the organization of morphologically distinct higher-order MT architectures in vitro. In cells, additional molecular factors play a role in modulating HSET activation and regulation. For instance, importin $\alpha/\beta$ is a demonstrated regulator of HSET activity in cells[47]. By binding to HSET's tail, importin $\alpha/\beta$ could act as a global on/off switch for HSET activity, thereby limiting the pool of tail domains available to bind either MT polymer or soluble (non-MT) tubulin. Additionally, HSET has been shown to bind EB1 and track the tips of growing MTs in vitro[48]. Thus, +TIP proteins may play an additional role in tuning HSET function at growing MT ends. Finally, eliminating the activity of HSET is sufficient to block the conversion of multipolar spindles to bipolar structures in cells with amplified centrosomes[11]. This has also been observed for dynein[49], another motor capable of forming MT asters in vitro. How a motor's ability to form MT asters in vitro translates to centrosome clustering in cells is currently unknown. Future studies determining the precise mechanisms of supernumerary centrosome clustering will be essential to understanding how cancer cells overcome this obstacle to continue their rapid cell division.

## Methods

**Construct design and sequence verification.** All constructs were prepared using an HSET cDNA corresponding to Gen Bank Accession Number BC121041. For motor domain (amino acids 305–673)-containing constructs, plasmids were prepared in the baculovirus vector pFASTBAC-HTa (Life Technologies) for use with the insect cell expression system, whereas all noncatalytic constructs (i.e., containing only the tail and stalk) were cloned into the bacterial expression vector pET15b (Novagen).

Amino acid sequences of the tail, stalk, and motor domains of HSET/KIFC1 (*Homo sapiens* kinesin-14) were determined by sequence alignment to Ncd (*Drosophila melanogaster* kinesin-14) using[24] as a guide, and used to design pET15b-HTa-EGFP-HSETΔMotor and pFASTBAC-HTa-EGFP-HSETΔTail. The pET15b-HTa-EGFP-HSETDoubleTail sequence was designed by sequence

alignment to XCTK2 (*X. laevis* kinesin-14) using[19] as a guide. Where desired, a flexible amino acid linker (GGSGGS) was inserted to ensure rotational freedom between domains.

Each construct contains the corresponding amino acid sequence, which was verified by DNA sequencing:

pFASTBAC-HTa-EGFP-HSET: MSYY-His$_6$-YDIPTTENLYFQGAMDPEF-EGFP(aa 1–239)-HSET(aa 1–673).

pET15b-HTa-EGFP-HSETΔMotor: MGSS-His$_6$-SSGLVPRGSH-EGFP(aa 1–239)-SGLRSRAQASNS-HSET(aa 1–304).

pET15b-HTa-EGFP-HSETDoubleTail: MGSS-His$_6$-SSGLVPRGSH-EGFP(aa 1–239)-SGLRSRAQASNS-HSET(aa 1–307)-GGSGGS-HSET(aa 1–137).

pFASTBAC-HTa-EGFP-HSETΔTail: MSYY-His$_6$-YDIPTTENLYFQGAMDPEF-EGFP(aa 1–239)-GGSGGS-HSET(aa 139–673).

pFASTBAC-HTa-HSET: MSYY-His$_6$-DYDIPTTENLYFQGAMDPEF-HSET(aa 1–673).

pEM791-EGFP-HSET: EGFP(aa 1-239)-HSET(aa1-673).

**Molecular biology and baculovirus construction.** All constructs (pFASTBAC-HTa-EGFP-HSET, pET15b-HTa-EGFP-HSETΔMotor, pET15b-HTa-EGFP-HSETDoubleTail, pFASTBAC-HTa-EGFP-HSETΔTail, pFASTBAC-HTa-HSET) were prepared by isothermal assembly[50]. PCR fragments consisting of HSET amino acids 1–673 (pFASTBAC-HTa-EGFP-HSET, pFASTBAC-HTa-HSET, pEM791-EGFP-HSET), 1–304 (pET15b-HTa-EGFP-HSETΔMotor), 139–673 (pFASTBAC-HTa-EGFP-HSETΔTail), and 1–307 + 1–137 (pET15b-HTa-EGFP-HSETDouble-Tail) were generated by amplification of the HSET ORF (using pEGFP-C1-HSET as a DNA template[37]) using Phusion DNA polymerase (New England Biolabs #M0530S). These fragments were then assembled into their expression vectors cut using the indicated restriction enzymes: pFASTBAC-HTa-EGFP-HSET: *EcoRI/XhoI*. pFASTBAC-HTa-EGFP-HSETΔTail and pFASTBAC-HTa-HSET: *EcoRI/KpnI*. pET15b-HTa-EGFP-HSETΔMotor and pET15b-HTa-EGFP-HSETDouble-Tail: *NdeI/XhoI*. pEM791-EGFP-HSET: *BsrGI/BglII*. Constructs in the pFASTBAC-HTa vector were used with the Bac-to-Bac system (Invitrogen) per manufacturers' protocol to create a baculovirus expressing the indicated protein.

The cloning of pFASTBAC-HTa-EGFP-HSET was described previously[37]. For other constructs, the following primers were used for isothermal assembly:

pET15b-HTa-EGFP-HSETΔMotor Forward Primer: "15b-_Cherry_IA_5":
5′-GCAGCGGCCTGGTGCCGCGCGGCAGCCATATGGTGAGCAAGGGCGAGG-3′

pET15b-HTa-EGFP-HSETΔMotor Reverse Primer: "HSET304REV":
5′-CGGGCTTTGTTAGCAGCCGGATCCTCGAGTCACAGCTGGTTGTGCAGTCG-3′

pET15b-HTa-EGFP-HSETDoubleTail Forward Primer 1: "15b-_Cherry_IA_5":

5′-GCAGCGGCCTGGTGCCGCGCGGCAGCCATATGGTGAGCAAGGGC
GAGG-3′

pET15b-HTa-EGFP-HSETDoubleTail Reverse Primer 1: "H137-GGS2-
H307REV":

5′-GACCTCTGCGGATCCATTGATCCTCCTGATCCTCCGAGTTCCT
GCAGCTGGTT-3′

pET15b-HTa-EGFP-HSETDoubleTail Forward Primer 2: "H307-GGS2-
H137FOR":

5′-AACCAGCTGCAGGAACTCGGAGGATCAGGAGGATCAATGGATCCG
CAGAGGTC-3′

pET15b-HTa-EGFP-HSETDoubleTail Reverse Primer 2: "15b-HSET137-REV":
5′-
CGGGCTTTGTTAGCAGCCGGATCCTCGAGTCACCAGGCTGGACGTTTGC
-3′

pFASTBAC-HTa-EGFP-HSETΔTail Forward Primer 1: "pFB-EGFP-FOR":
5′-GTATTTTCAGGGCGCCATGGATCCGGAATTCATGGTGAGCAAGGG
CGAGG-3'

pFASTBAC-HTa-EGFP-HSETΔTail Reverse Primer 1: "H139-GGS2-EGFP-
REV":

5′-GTCACATAACTGACCCTTTAATGATCCTCCTGATCCTCCCTTGT
ACAGCTCGTCCATG-3′

pFASTBAC-HTa-EGFP-HSETΔTail Forward Primer 2: "H139-GGS2-H673-
FOR"

5′-CATGGACGAGCTGTACAAGGGAGGATCAGGAGGATCATTAAAGGG
TCAGTTATGTGAC-3′

pFASTBAC-HTa-EGFP-HSETΔTail Reverse Primer 2: "pFB-HSET_UnlREV":
5′-TCCTCTAGTACTTCTCGACAAGCTTGGTACCTCACTTCCTGTTGG
CCTG-3′

pFASTBAC-HTa-HSET Forward Primer: "pFB-HSET_UnlFOR":
5′-GTATTTTCAGGGCGCCATGGATCCGGAATTCATGGATCCGCAGAG
GTC-3′

pFASTBAC-HTa-HSET Reverse Primer: "pFB-HSET_UnlREV":
5′-TCCTCTAGTACTTCTCGACAAGCTTGGTACCTCACTTCCTGTTGGC
CTG-3′

pEM791-EGFP-HSET Forward Primer: "EM791_GFP-HSET_5′":
5′-CACTCTCGGCATGGACGAGCTGTACAAGATGGATCCGCAGAGG
TC-3′

pEM791-EGFP-HSET Reverse Primer: "EM791_GFP-HSET_3′":
5′-CATAATTTTTGGCAGAGGGAAAAAGATCTTCACTTCCTGTTGGC
CTG-3′

**Protein expression and purification.** Constructs in the pFASTBAC-HTa vector
were expressed in *Sf*9 cells (Invitrogen) by infecting them with the corresponding
baculovirus for 72 h. Constructs in the pET15b vector were expressed in BL21DE3
cells (Stratagene) with 0.4 mM IPTG for 16 h at 16 °C. Each protein was purified
via His$_6$-affinity chromatography, followed by size exclusion chromatography, as
follows: cells were pelleted and resuspended in PNI buffer (50 mM sodium phos-
phate, 500 mM NaCl, 20 mM imidazole) with 5 mM β-mercaptoethanol (β-ME),
1% NP40, and protease inhibitors (1 mM phenylmethylsulfonyl fluoride, 1 mM
benzamidine, and 10 µg/mL each of leupeptin, pepstatin, and chymostatin). Bac-
terial cells were incubated on ice with 1 mg/mL lysozyme (Sigma) for 30 min to
remove bacterial cell walls; this step was omitted for *Sf*9 purifications. Cells were
then lysed by sonication and clarified by centrifugation at 4 °C at 35 K RPM
(142,414×*g*) for 1 h in a Ti-45 rotor (Beckman). 4 mL of Ni$^{++}$-NTA agarose
(Qiagen) were incubated with the supernatant for 1–2 h, then washed extensively
with wash buffer (PNI, 5 mM β-ME) for 3–4 h. For proteins containing the cata-
lytic motor domain, this wash buffer was supplemented with 50 µM MgATP to
ensure proper nucleotide binding to the motor. The proteins were then eluted from
the Ni$^{++}$-NTA agarose column in PNI with 5 mM β-ME and 180 mM imidazole
(supplemented with 100 µM MgATP for proteins containing the motor domain).
Peak fractions (5 mL total) were then subjected to size exclusion chromatography
on a Hiload 16/60 Superdex 200 preparatory grade column (GE Healthcare) in gel
filtration buffer (10 mM K-HEPES, pH 7.7, 1 mM DTT, 300 mM KCl, supple-
mented with 100 µM MgATP for proteins containing the motor domain). Protein
concentrations were determined in mg/mL using Bradford assays (BioRad) and
converted to molar units assuming dimer formation. Powdered sucrose was added
to 20% w/v, then each protein was aliquoted, snap frozen in liquid nitrogen, and
stored at −80 °C.

**Tubulin purification and labeling.** Bovine brain tubulin was purified according to
Castoldi and Papov[51]. Briefly, brain homogenates were cycled twice in high-
molarity PIPES buffer (1 mM K-PIPES, pH 6.9, 10 mM MgCl$_2$, 20 mM EGTA) to
remove MT-associated proteins, resuspended in BRB80 buffer (80 mM K-PIPES,
pH 6.9, 1 mM MgCl$_2$, 1 mM EGTA), diluted to 20 mg/mL, snap frozen in liquid
nitrogen, and stored at −80 °C. To generate fluorescently labeled or biotinylated
tubulin, NHS-Alexa594, NHS-Cy5, and NHS-Biotin (Thermo Fisher #A-20104,
#A-20106, and #20217, respectively) were conjugated to cycled tubulin using
succinimidyl ester chemistry essentially as described[52]. Briefly, 10 mg/mL tubulin
was polymerized in BRB80 supplemented with 50% glycerol and 1 mM GTP for
30 min at 37 °C, then spun over a high pH (pH 8.6) glycerol cushion at 40 K RPM

(193,357×*g*) for 45 min in a Ti-50.2 rotor (Beckman) at 37 °C. Polymerized tubulin
was washed 2×, the dye (indicated above) was added at 20× molar excess, and
incubated at 37 °C for 40 min. The reaction was quenched in 50 mM K-glutamate
for 5 min, then pelleted by spinning over a low-pH (pH 6.9) glycerol cushion at
80 K RPM (346,214×*g*) for 20 min in a TLA 100.3 rotor (Beckman) at 37 °C.
Labeled pellets were depolymerized by resuspension in ice-cold 50 mM K-
glutamate +0.5 mM MgCl$_2$ (pH 7.0) and subjected to continuous dounce homo-
genization for 25 min on ice. The labeled soluble tubulin was clarified by spinning
at 80 K RPM (346,214×*g*) for 10 min in a TLA 100.3 rotor (Beckman) at 4 °C. The
clarified supernatant was then repolymerized in BRB80 supplemented with 50%
glycerol and 1 mM GTP at 37 °C for 30 min. The polymerized, labeled mixture was
spun again over a low-pH glycerol cushion and depolymerized as above. After a
final clarifying spin (see above), the supernatant was recovered and flash frozen in
2 µL aliquots for subsequent use. In TIRF experiments with Cy5-labeled tubulin,
100% labeled tubulin was used.

**Self-organization assays.** Self-organization experiments were performed similar
to previous studies[19]. Experiments were performed in narrow (~10 µL volume)
flow cells prepared by attaching a clean #1.5 coverslip (Fisherbrand) to a glass slide
(Thermo) with double-sided tape (Scotch). For all experiments, the experimental
mixtures were assembled at room temperature (rather than on ice) to prevent
GMPCPP-MT depolymerization. Tubulin, GMPCPP-MTs, and the indicated
HSET constructs were added to a 4× self-organization buffer, which was then
diluted with water to the following final concentrations: 20 mM K-PIPES, pH
6.8, 1 mM EGTA, 7.5 mM MgCl$_2$, 5 mM ATP, 1.5 mM GTP, 50 mM KCl, 200 mM
sucrose, 2 µM taxol, 250 µg/mL casein, and an oxygen scavenging mix (200 µg/mL
glucose oxidase, 35 µg/mL catalase, 25 mM glucose, and 70 mM β-ME). Tubulin
and HSET were added at the concentrations indicated in figure legends. GMPCPP-
MTs were prepared as follows: 20 µM bovine brain tubulin (10% Alexa594-
labeled), 1 mM GMPCPP, and 1 mM DTT were mixed in 1× BRB80 and incubated
on ice for 5 min, then transferred to 37 °C for 1 h. When present, 1 µL GMPCPP-
MTs were used in a total volume of 20 µL experimental volume for a theoretical
polymer concentration of 1 µM. In all experiments, HSET was added last to
minimize premature cross-linking of MTs. The chamber was loaded with self-
organization mix, sealed with VALAP, and immediately (<1 min) mounted to a
preheated microscope chamber at 37 °C.

Imaging was performed at 37 °C on an inverted DeltaVision Elite (GE
Healthcare) microscope equipped with a WeatherStation environmental chamber
(Applied Precision), a 20× lens (NA = 0.75) (Olympus), and a CoolSnapHQ2 CCD
camera (Roper), and controlled by SoftWorx image acquisition software (GE
Healthcare). Time-lapse wide-field fluorescence images were acquired using
standard TRITC and FITC filters at 15 s intervals for a total duration of 30 min and
the focus was fine-tuned as necessary. Contrast was defined as the standard
deviation of the entire field of view measured in the tubulin channel, and was
calculated using ImageJ.

**TIRF experiments for motility and single-molecule behavior.** All TIRF assays
were performed in narrow (~10 µL volume) flow cells prepared by attaching a clean
#1.5 coverslip (Fisherbrand) to a glass slide (Thermo) with double-sided tape
(Scotch). Flow chambers were processed at room temperature by infusing with
biotinylated bovine serum albumin (BSA) (Thermo Fisher Pierce #29130) at 2 mg/
mL in BRB80 for 10 min, washing twice in BRB80 supplemented with 0.5 mg/mL
casein (BRB80/casein), infusing with NeutrAvidin (Invitrogen #31000) at 1.67 mg/
mL in BRB80 for 10 min, washing twice in BRB80/casein, and blocking with 1%
Pluronic F-127 (Sigma #P2443) in BRB80 for 20 min. After two additional washes
in BRB80/casein, fluorescent, biotinylated GMPCPP-MTs (prepared as above; 70%
unlabeled, 20% biotinylated, 10% Alexa594-labeled) were introduced at a 1:40
dilution in BRB80 and incubated for 2 min to adhere to the surface, and the flow
chambers were washed twice in BRB80/casein. For single-molecule diffusion
experiments, the final experimental mix was assembled in P12 buffer (12 mM
PIPES/KOH, pH 6.8, 1 mM EGTA, and 2 mM MgCl$_2$) containing 1 mM ATP,
0.5 mg/mL casein, an oxygen-scavenging mix (see above), and EGFP-HSET at the
concentrations indicated in the figure legends. For all other experiments studying
motility behavior, the final experimental mix was assembled in BRB80 buffer
containing 50 mM KCl, 1 mM ATP, 1 mM MgCl$_2$, 0.5 mg/mL casein, an oxygen-
scavenging mix (see above), and HSET/tubulin at the concentrations indicated in
the figure legends. The chamber was loaded with experimental mix, sealed with
VALAP, and immediately (<1 min) mounted to the TIRF microscope with
objective heater (Tokai Hit) set to 35 °C.

Imaging was performed at 35 °C using an inverted Ti-E microscope (Nikon)
equipped with an H-TIRF module, Perfect Focus, 100× Apochromat total internal
reflection fluorescence (TIRF) objective (NA 1.49, Nikon) with a 1.5× tube lens,
EMCCD detector (iXon Ultra DU897; Andor Technology), a LuNA solid state laser
system with four laser lines (405 nm, 488 nm, 561 nm, and 640 nm, ~10 mW power
at output) combined into a single fiber and rapidly controlled with an acousto-
optic tunable filter (AOTF, Nikon). Using Nikon Elements image acquisition
software, images were acquired continuously with 100 ms exposures for 1 min
(single-molecule diffusion experiments) or 1 s exposures for 10–15 min (processive
motility experiments), and image acquisition was controlled by Nikon Elements
software. For near-simultaneous two-color imaging of HSET and tubulin, a quad-

band filter turret was used (C-FL TIRF Ultra Hi S/N 405/488/561/638 Quad Cube), and the AOTF was used to rapidly switch between 488 nm and 640 nm laser excitation with 100 ms exposures in each channel.

**Single-molecule particle tracking and MSD analysis**. For single-molecule particle tracking, the SpotTracker plugin for ImageJ[53] [http://bigwww.epfl.ch/sage/soft/spottracker] was modified to batch-process motility data, and the frame-by-frame centroid position of each particle was determined as described[54]. Briefly, particles above a brightness threshold were detected automatically and the centroid position was determined by fit to a 2D Gaussian function to determine the frame-by-frame position with subpixel resolution. Manual detection was used when photoblinking or nearby particle(s) obstructed automatic detection. Particles persisting at least 5 frames (500 ms) were considered, and particle tracking ended when particles intersected a neighboring particle or disappeared completely (either by photobleaching or unbinding from the MT). Diffusion coefficients were calculated using the MSD analyzer MATLAB plugin[55], where the first 5 s were used for MSD analysis.

**Kymograph and motility analysis**. Processive motility properties (i.e., end dwell times, run lengths, velocities, and event frequencies) were analyzed by kymograph. Maximum intensity projections were generated to determine the location of MTs in the channel of interest, and kymographs were generated using the Multiple Kymograph plugin in ImageJ (width = 3 pixels). End dwell time was defined as the vertical component of the kymograph, which is the time spent at the end of the MT, in seconds. For each end dwell event in the green channel, the particle was compared to the MT location in the red channel to verify its location at the end of the MT rather than along the lattice. For a particle to be considered for analysis, it: (i) must persist at least 5 frames (5 s), (ii) could not coincide with other particles at the end of the MT, and (iii) must arrive and detach at the end of the MT during image acquisition. Run length was defined as the horizontal component of the kymograph, which is the distance traveled along the MT in μm, and included all pauses. For a particle to be considered for run length analysis, it must: (i) persist at least 5 frames (5 s), (ii) begin its run during image acquisition, and (iii) must not be bound to the MT at the end of image acquisition. Events that reached the end of the MT were counted in the analysis but treated separately (as indicated in the figure legends). Velocity was defined as the run length (horizontal component) divided by time (vertical component) in μm/s. Only moving segments were considered, i.e., if no movement was observed in a segment at least 5 frames (5 s) long, this portion was excluded from the analysis. For a particle to be included for velocity analysis, it: (i) must persist at least 5 frames (5 s), (ii) begin its run during image acquisition, and (iii) must not be bound to the MT at the end of image acquisition. For event frequency measurements (Events/(min*μm)), time was defined as the image acquisition time (min), MT length was determined from the red channel (μm), and events were determined by kymograph analysis, defined as moving (nonstationary) particles persisting at least 5 frames (5 s).

**Co-immunoprecipitation assays**. For co-immunoprecipitation, purified components (concentrations: 50 nM for the indicated EGFP-HSET truncation and 250 nM for tubulin) were mixed to a total volume of 190 μL on ice in BRB80 buffer containing 50 mM KCl, 1 mM ATP, 1 mM GTP, 2 mM MgCl₂, and 0.5 mg/mL casein. A 10 μL of triply washed anti-GFP Binding Magnetic Beads (Vanderbilt Antibody and Protein Resource, Nashville, TN) were then added and the solution was allowed to rotate overnight at 4 °C. Beads were pelleted by a 5 min incubation on a magnetic rack, washed twice in 500 μL BRB80, resuspended in 20 μL, and diluted 1:1 in 2× sample buffer (100 mM Tris-Cl, pH 6.8, 4% SDS, 20% glycerol, 200 mM DTT, and 200 μg/mL bromophenol blue), and boiled. Samples were then resolved by SDS-PAGE on a polyacrylamide (10%) gel and transferred to a nitrocellulose membrane for immunoblotting. Immunoblots were analyzed by blotting with a monoclonal mouse antibody to anti-His₆ at a 1:1500 dilution (GE, #27-4310-01) and a polyclonal rabbit antibody to alpha-tubulin at a 1:5000 dilution (Abcam #18251). Inputs were loaded at 15% of the total molar amount of the immuno-precipitated material.

**Fluorescence intensity analysis**. To determine single-molecule fluorescence intensities for EGFP-HSET and Cy5-tubulin, 50 pM of material was introduced to an unblocked narrow flow cell (~10 μL volume) to nonspecifically adhere single molecules to the glass surface similar to previous studies[30]. Samples were diluted in BRB80 supplemented with 0.5 mg/mL casein for 2 min. Flow cells were then washed twice with 50 μL BRB80 supplemented with 0.5 mg/mL casein to remove nonadsorbed fluorescent proteins, and imaged by TIRF (see above). These particles were compared to the first frame of moving fluorescent particles in TIRF-based motility assays on the same day using identical imaging and analysis parameters.

To obtain first frame intensity values, a 2D Gaussian fitting routine was implemented in ImageJ (http://www.sussex.ac.uk/gdsc/intranet/microscopy/imagej/gdsc_plugins#install), local maxima were detected automatically, and the reported Gaussian intensity values (area under the curve) were reported after local background correction. For time-lapse intensity traces showing stepwise photobleaching, background-corrected intensity values were obtained by implementing the 2D Gaussian fitting routine on the same x–y position in time.

The intensity was reported as zero when the intensity fell below the required threshold for the fitting routine.

**Statistical analysis and error determination**. Wherever mean ± 95% CI was reported, cumulative distribution functions (CDFs) of distributions were generated in MATLAB and mean values were obtained similar to previous studies[56]. CDFs were used for statistical analysis because they are continuous and do not introduce subjective binning. Using MATLAB, fluorescence intensity and velocity CDFs were fit to the hypothetical CDF for a normal distribution using a nonlinear least-squares fit with free parameters $\mu$ (mean) and $\sigma$ (standard deviation, SD):

$$CDF(x) = \frac{1}{2}\left[1 + erf\left(\frac{x-\mu}{\sqrt{2\sigma^2}}\right)\right] \quad (1)$$

Errors were obtained by the bootstrap technique, where each distribution as resampled 200 times and fit to the normal distribution. The SD of the mean parameter $\mu$ over the resampled datasets was taken as the standard error for each fitted quantity, and the data were reported as the mean ± the 95% CI, where the 95% CI was defined as two times the standard error.

Similar to the normal distributions above, run length CDFs above a minimum $x_0 = 0.3$ μm were fit to the hypothetical CDF for an exponential distribution with the free parameter $t$:

$$CDF(x) = 1 - e^{(x0-x)/t} \quad (2)$$

The mean run length was then determined by adding the minimum run length to $t$. End dwell CDFs were fit to the same hypothetical exponential CDF without the minimum correction. Errors for these values were then determined as above. For display, Gaussian or exponential functions using the determined parameters were overlaid to the binned data in Excel.

**Analytical ultracentrifugation**. Purified tubulin (5 μM) with or without EGFP-HSET (200 nM) in BRB80 was analyzed in an Optima XLI ultracentrifuge (Beckman Coulter, Brea, CA) equipped with a four hole An-60 Ti rotor at 42,000 RPM (142,249×g) at 4 °C, where absorbance at 280 nm was monitored. Samples were loaded into double-sector cells (path length of 1.2 cm) with charcoal-filled Epon centerpieces and sapphire windows. Sedfit (version 15.0) was used to analyze velocity scans using every scan from a total of between 250 and 300 scans[57]. Approximate size distributions were determined for a confidence level of $p = 0.95$, a resolution of $n = 300$, and sedimentation coefficients between 0 and 30 S. For tubulin and tubulin with EGFP-HSET, the frictional ratio was allowed to float.

**Quantum dot assembly**. For experiments using QDots, HSET/QDot assemblies were constructed as follows. A biotinylated mouse monoclonal anti-His₆ antibody (final concentration 0.1 mg/mL, Invitrogen #MA1-21315-BTIN) was mixed with streptavidin-coated QDots (final concentration 600 nM, QDot 655 streptavidin conjugate, Molecular Probes #Q10123MP) in BRB80 and incubated overnight at 4 °C to create anti-His QDots. The following day, anti-His QDots (final concentration 20 nM) were mixed with the indicated HSET construct (final concentration 60 nM) and incubated for at least 30 min on ice to form HSET/QDot assemblies at a 3:1 ratio. For TIRF experiments where biotinylated GMPCPP-MT seeds were used, any unbound streptavidin on the QDot surface was quenched by incubating HSET/QDot assemblies (final concentration 15/5 nM) with D-Biotin (ACROS Organics #230095000) in BRB80 for at least 10 min on ice. All experiments were then performed as described above using the indicated concentration of HSET/QDot assemblies.

**Stable cell line generation**. To generate HeLa-Kyoto cells (Naoki Watanabe) that express EGFP-HSET in a doxycycline-inducible manner, HeLa acceptor cells (described previously[30]) were transfected in six-well plates with 1 μg/μL of pEM791-EGFP-HSET. One day after transfection, cells were cultured in the presence of 1 μg/mL puromycin for 48 h and then incubated in media containing 2 μg/mL puromycin until puromycin-sensitive cells were eliminated. Puromycin-resistant cells were then expanded in media containing 1 μg/mL puromycin, pooled, and tested negative for mycoplasma contamination. Cells were cultured in DMEM containing 10% FBS, 1 μg/mL puromycin, penicillin/streptomycin, and either 2 μg/mL doxycycline (to induce EGFP-HSET expression) or DMSO for 1–3 d.

**Cell lysis and western blotting**. Doxycycline/DMSO-treated cells were cultured in six-well plates until reaching ~80% confluency. Cells were trypsinized and harvested by low-speed centrifugation at 1500×g at 4 °C. The pellet was washed once in DMEM and resuspended in 50 μL lysis buffer (25 mM HEPES/KOH, 115 mM potassium acetate, 5 mM sodium acetate, 5 mM MgCl₂, 0.5 mM EGTA, and 1% Triton X-100, pH 7.4) freshly supplemented with 1 mM ATP and protease inhibitors (1 mM phenylmethylsulfonyl fluoride, 1 mM benzamidine, and 10 μg/mL each of leupeptin, pepstatin, and chymostatin). After the lysate was clarified by centrifugation at 16,000×g at 4 °C, samples were loaded directly onto a 10%

polyacrylamide gel at 1/2 well per sample. Samples were resolved by SDS-PAGE and transferred to a nitrocellulose membrane for immunoblotting.

For immunoblotting, nitrocellulose membranes were blocked in 5% milk diluted in Tris-buffered saline plus 0.1% Triton X-100 (TBST). Rinses were performed in TBST. Primary antibodies were diluted in 5% milk/TBST and probed for 1 h. The following primary antibodies were used for immunoblotting: mouse anti-tubulin (DM1α, Vanderbilt Antibody and Protein Resource), 1:2000, rabbit anti-GFP (Invitrogen #A11122), 1:1000, and rabbit anti-HSET (affinity-purified from polyclonal serum, a generous gift from Dr. Duane Compton), 1:2000. Species-specific secondary antibodies conjugated to Alexa Fluor 700/800 (Invitrogen) were used at 1:5000 for 45 min. Bound antibodies were detected using an Odyssey fluorescence detection system (Mandel Scientific). Protein levels were quantified with Fiji using tubulin as a loading control.

**Immunofluorescence imaging and quantification.** Doxycycline/DMSO-treated cells were cultured in six-well plates until reaching ~80% confluency, then treated with 500 nM nocodazole (to partially depolymerize MTs) or DMSO for 15 or 30 min. Immediately following nocodazole treatment, cells were fixed with methanol at −20 °C for 10 min. Cells were rehydrated with 3 × 5 min washes with TBST. Coverslips were blocked with AbDil (TBST + 2% BSA [Sigma]) for 10 min, probed with primary antibodies diluted in AbDil for 1 h, rinsed, probed with secondary antibodies diluted in AbDil for 1 h, and rinsed. The following primary antibodies were used for immunostaining: mouse anti-tubulin (DM1α, Vanderbilt Antibody and Protein Resource), 1:500, rabbit anticentrin (preparation described previously[58]), 1:1500. DNA was stained with 5 μg/mL Hoechst 33342. Coverslips were mounted in Prolong Gold for imaging.

Images in four channels (DNA, EGFP-HSET, tubulin, and centrin) were acquired at 37 °C on an inverted DeltaVision Elite (GE Healthcare) microscope equipped with a WeatherStation environmental chamber (Applied Precision), a 60× objective (NA = 1.4) (Olympus), and a CoolSnapHQ2 CCD camera (Roper). Z-sections spaced at 200 nm apart were acquired and deconvolved with SoftWorx (GE Healthcare). Images were subsequently processed with Fiji (maximum intensity Z-projections, adjusting minimum and maximum levels, rotating, and cropping).

To determine the fraction of mitotic cells that had ≥1 acentrosomal aster after nocodazole treatment, metaphase cells were selected for imaging and analysis using the blue channel (DNA). For every cell, each discrete mass of tubulin was manually evaluated for colocalization with centrin using deconvolved Z sections of tubulin and centrin. An acentrosomal aster was classified as a discrete polymer of tubulin lacking a centrin marker.

**Live-cell imaging.** For live-cell imaging of aster formation, doxycycline/DMSO was added to EGFP-HSET cells as they were plated onto glass bottom poly-D-lysine coated dishes (MatTek) 3 d before imaging. Cells were imaged at 37 °C (5% CO$_2$) in L-15 medium without phenol red supplemented with 10% FBS and 7 mM HEPES, pH 7.7, using the aforementioned DeltaVision Elite system. Metaphase cells were identified in the EGFP-HSET channel, whereupon 500 nM nocodazole was added to partially depolymerize MTs. Immediately (~10 s) after nocodazole addition, Z-sections spaced at 200 nM apart were acquired in the EGFP-HSET channel, and Z-stacks were acquired at 3 min intervals for 30 min.

**Data availability.** Data supporting the findings of this manuscript are available from the corresponding authors upon reasonable request.

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

## Acknowledgments

We thank members of the Ohi and Zanic laboratories for critical insights, and we thank Drs. Matt Lang and Kristen Verhey for valuable discussion and feedback. We thank Dr. Duane Compton for the generous gift of the anti-HSET antibody. This work was supported by National Institutes of Health Grants R01GM066610 to R. Ohi and R35GM119552 to M. Zanic. R. Ohi is a Scholar of the Leukemia and Lymphoma Society. M. Zanic acknowledges the support from the Human Frontier Science Program and the Searle Scholars Program. C. Strothman is supported by NIH Grant 5T32GM008554-21, and A. Erwin is supported by NIH Grant 2T32CA119925-09. We thank the Vanderbilt Center for Structural Biology (CSB) for providing funds that maintain the analytical ultracentrifuge used in this study.

## Author contributions

S.N., M.O., M.Z., and R.O. designed research; S.N., S.J., P.S., C.S., A.E., and R.O. performed research; S.N., S.J., and P.S. analyzed data; and S.N., M.Z., and R.O. wrote the manuscript.

## Additional information

**Competing interests:** The authors declare no competing interests.

