## [Peer Review File · Nature Communications]

Reviewers' comments:

Reviewer #1 (Remarks to the Author):

The manuscript by Norris et al. describes the surprising finding that the human kinesin-14 protein HSET can switch from a non-processive to processive mode of movement along MTs depending on the concentration of soluble tubulin in solution. This switch allows HSET to transition from a potent MT crosslinking motor that can slide anti-parallel MT bundles, a function previously well-characterized for class 14 kinesins, to one that can form stabilized MT asters in vitro. The proposed mechanism of this switch is the clustering of multiple kinesin motors together through interactions with soluble tubulin, similar to recently proposed motor clustering mechanisms for other kinesin-14 family members in diverse species. This discovery, further opens the broader question of what other co-factors play a role in molecular movement in cells. The results are very interesting and informative for the motor protein field, and should be of interest to broader audiences. The experiments appear to be of high technical quality. I support publication after the authors address the following concerns:

Major Concerns:

1) The crux of the paper is the novel finding of HSET-tubulin multimers that move processively to minus-ends. The data are convincing that this phenomenon is happening in the author's experiments. However, the authors appear to gloss over the nature of these multimers in the text. They go so far as to quantify fluorescence intensities of both HSET and tubulin in these multimers, but fail to put a final number of each molecule on the moving particles (3 HSET's and 12 tubulins?). I'm guessing because of the broad range of intensities they observe? The histograms of intensities for both HSET and tubulin show a distinct peak within the range of the single molecule intensities (Fig. 3 and S3) and the fits are not convincing for single Gaussian. The authors should provide R² values for the fits and more importantly try fitting with multiple Gaussians. How do the authors explain the processive particles that have the same intensity as single HSET molecules? I think the molecular nature of the HSET-tubulin oligomer requires much more discussion in the manuscript.

2) The bigger point is can the authors provide more robust characterization of these particles to make a more convincing argument that what they are observing is a bona-fide molecular structure versus and in vitro artifact? Having done a lot of single molecule TIRF myself, I can say for certainty that it is very easy to find movement of aggregates within a particular prep of motor molecules which should not normally move on their own (monomers for instance). Further, tubulin is known to be sensitive to aggregation, and the addition of tubulin may introduce aggregates that nucleate motor clusters. Have the authors performed high-speed ultracentrifugation or gel filtration of their proteins (particularly tubulin) just prior to the TIRF assay as a control for aggregation-induced artifacts?

3) The IP experiment is a decent start to show that the tail of HSET can bind tubulin, but can the authors isolate a stable HSET-tubulin co-complex by gel filtration or sucrose gradient sedimentation? It would be much more convincing if the authors could demonstrate a stable co-complex of defined molecular composition (SEC-MALS or AUC?). Or do the authors think that moving particles are heterogeneous in numbers of motors/tubulins, and that is the nature of the mechanism in cells? If so, this point should be raised in the discussion.

4) I think Fig. 3 could be revamped. I don't find the images of particles in B and C very enlightening. I would replace the particle images with the data from Fig. S3B, C. I'm also confused how they got the brightness measurements at 10nM Cy5-tubulin. Is 10nM the total tubulin concentration in this experiment, which seems oddly low since all the other processive movement is seen at much higher tubulin concentrations? I have the same question for Fig. 2G and Fig. S3C. The efficiency of tubulin labeling should be reported as well, as this is critical for the calculation of

how many tubulins are in a given spot. Have the authors taken this into account?

5) The self-organization assay is convincing and a nice addition that reconciles past observations. How do the authors envision that HSET "toggles" between modes of motility in vivo given the high intracellular concentration of soluble tubulin? A sentence or two on this point seems warranted.

Minor Points:

1.) Fig. S1. Two-step photobleaching is a good start towards demonstrating the oligomeric state of the HSET preparation. However no statistics are given about the data. What fraction of motor spots showed two-step vs. other steps of bleaching, and how many spots were counted/excluded. In my experience, step photobleaching can be tricky and not very robust. It would be good if the authors had another confirmation of their protein state (intensity analysis compared to another dimeric, GFP-tagged protein?).

2.) Fig. 3B, C. I was surprised that the authors do not put a final estimated number of molecules on their quantifications of motor and tubulin intensities, even after starting the text section by stating that they counted numbers of molecules.

3.) Although importin α/β is a likely regulator of HSET via binding to its tail domain (38), this mechanism would more likely act as a global on/off switch for HSET activity rather than toggling between MT sliding and processive motion. The authors should expand on this point as it seems very relevant to their mechanism.

4.) Fig. 2H is mislabeled

5.) The authors claim to visualize HSET and tubulin "simultaneously" in Fig. 2G and Movie 5. Is this true simultaneous imaging? If so, the authors have failed to describe the microscope setup for imaging two channels simultaneously (i.e. are they using an image splitting device or two cameras).

6.) The authors claim that there are approximately three HSET-FL molecules per Qdot in their experiments, but the numbers in Fig. S4 look more like two motors per Qdot.

7.) The tubulin channel in movie S4 seems strangely cropped compared to the HSET channel in the 1nM concentration panel.

8.) The authors make the point to mention that the fraction of moving motors increased as concentration of HSET increased. Do they suggest cooperativity or interactions between HSET molecules?

9.) The authors report dwell time data at minus-ends with Qdots only. It is not clear what the point of this data is in the current manuscript. I would recommend performing dwell time analysis for native clusters. If the numbers are similar, it is an additional parameter to show Qdots are effectively mimicking the tubulin induced clustering of motors.

Reviewer #2 (Remarks to the Author):

In this work, the authors study the formation of microtubule asters by the human kinesin-14, HSET. They found that HSET can form asters from dynamic microtubules, but cannot form asters from stabilized (i.e. non-dynamic) microtubules. They explain this finding by the fact that HSET, a non-processive motor, becomes processive in the presence of soluble tubulin through the formation of multi-motor tubulin-HSET clusters.

The transition from the non-processive state to the processive might be an interesting switch mechanism, the authors however at the moment fail to present convincing evidence that the clustering, which drives the switch, is not a result of rather non-specific protein aggregation. Any aggregates of a non-processive motor are likely to become processive. To rule this out, the authors should investigate the clusters further, for example by i) showing in detail how are these clusters assembled (e.g. by biochemical mapping of the interaction interfaces going beyond the current identification of the HSET tail-domain), ii) investigating the structure of the clusters (e.g. by electron microscopy), and/or iii) showing that similar clustering occurs in the cell. Moreover, aster formation of dynamic microtubules by kinesin-14 has already been shown by Hentrich and Surrey for XCTK-2 (ref. 1 in the manuscript). The additional insight provided by the current manuscript is rather marginal, given that no mechanistic understanding of the cluster formation (beyond the purple ellipses in Fig. 5B) is offered. I can therefore not recommend consideration for publication in Nature Communications.

Specific comments:

1) The authors say that "These particles are unlikely to be aggregates, because: (i) moving particles appeared as diffraction-limited spots... (iii) the intensity distributions were still well-fit to a normal distribution..."

Neither seems to be the case, see figs. 3B,C

2) The authors never comment on the structure of the clusters. Does the HSET tail bind to several tubulin dimers? If so, why wouldn't the clusters grow further with increasing concentration of HSET/tubulin? In fact, it would not surprise me, if a similar cluster generation and microtubule aster formation would also occur in the presence of other negatively charged proteins (e.g. free globular actin instead of tubulin).

3) The ionic strength of the measurement buffer is known to affect protein aggregation. Why were the experiments performed in three different buffers / three different ionic strengths?

- self-organization in 20 mM K-PIPES, 50 mM KCl
- single-molecule diffusion experiments 12 mM PIPES
- all other experiments studying motility behavior BRB80 buffer containing 50 mM KCl

4) Minor: Figure "2H" should probably be "2G".

Reviewer #3 (Remarks to the Author):

In this manuscript the authors investigate the role of the kinesin14, HSET, in forming microtubule asters in vitro. The data nicely show that in single molecule assays, HSET is non-processive; that adding tubulin to HSET increases processivity; that tubulin co-localizes with processive motors; that adding tubulin to HSET and stable microtubules can drive aster formation; and that HSET on a Q-dot mixed with stable microtubules also results in aster formation.

My main concern is the nature of the tubulin that interacts with the motors and the stoichiometry of the motor tubulin complex. In the experiment in which Qdots were used there is no soluble tubulin: the motors are linked to the Q-dots via the streptavidin-biotin-his tag on the motor and added to stabilized microtubules. Under these conditions, asters form. This result suggests that HSET tail can bind to a cargo microtubule because tailless HSET on the Q-dot fails to drive asters from stable microtubules. However, other experiments show that GTP is not needed for inducing aster formation from a mixture of motors, stable microtubules and tubulin, suggesting that it is

soluble dimers, not microtubules, that are needed to bind to the motor, induce clustering. Is the GTP totally absent from the solution in this experiment? Do the authors envision that one dimer binds to two (or more) motors, generating a cluster? It would seem more likely that the soluble tubulin forms a short oligomer or protofilament of some sort, which could then bind to more than one motor, generating a cluster. One control experiment comes to mind: adding colchicine to the tubulin will prevent oligomerization. Does tubulin-colchicine, HSET and stable microtubules result in aster formation? In many places in the manuscript the authors state that HSET tail binds soluble tubulin, but is soluble tubulin responsible for clusterformation?

Related to this concern about tubulin vs a microtubule functioning to form clusters, is the data in Figure 3. The distribution of fluorescence for the motile motors and tubulin shows a very wide range of values. It is not clear if there is some preferred stoichiometry, nor is it clear if some of these are aggregates (even multiple molecules, of say 8nm each, can be diffraction limited spot). And if there are indeed 'aggregates' or oligomers, maybe this is required to induce clustering. This experiment was done with a relatively low concentration of HSET; what happens at higher concentrations of motor, or lower concentrations of tubulin – does the distribution of fluorescence values shift?

The authors need to show if soluble tubulin in a solution that has no GTP or that has colchicine, is sufficient for aster formation or if the tubulin is forming a higher order structure which then clusters the motors.

Other concerns:

The data in Figure 2 show that processivity is enhanced with higher concentrations of HSET, but only 10nM and 20nM are shown; what happens at 15nM? It seems important to include another concentration to learn if there is a linear relationship for this experiment or not.

When discussing adding a Q-dot to HSET, the text does not mention that the Qdot is a Streptavidin tagged etc. The text should include this information.

Figure legend does not mention panel H.

Figure 3 has no molecular weight markers on the blot.

The manuscript refers to asters and bundles of microtubules as "structural motifs". I do not think that this is an accurate use of that term. Could the authors please consider describing the micron scale organization of microtubules in a different way, for examples, 'microtubules can be organized into different architectures', or into 'different micron scaled structures', or simply that microtubules can be organized into bundles and asters.

I like the term 'cluster' not 'team'; the latter sounds like draught animals or a sports metaphor. Please switch to cluster.

After synthesizing reviewer comments, we felt that reviewer comments from all three reviewers fall under three major or general categories. Before addressing reviewer comments on a point-by-point basis, we wanted to share an overview of our efforts to revise the manuscript.

1. Concerns related to protein aggregation. As pointed out by the editor, a major concern shared by each reviewer was that clustering may be the result of non-specific protein aggregation in our *in vitro* assays. We have addressed this in two major ways. First, we incorporated suggestions from reviewers to rule out the presence of tubulin or HSET aggregates (e.g., analytical centrifugation, using SEC-purified tubulin to activate HSET, expanded stepwise photobleaching analysis) in our *in vitro* assays. Second, we are excited to present evidence that our findings *in vitro* are very likely to be relevant in cells. In brief, we demonstrate that soluble tubulin activates HSET in cells. The most important contribution of this new data set is that it will help to alleviate any concern that tubulin-mediated HSET motor activation is an *in vitro* artifact.

2. Composition of HSET-tubulin clusters. This area of concern centered on the stoichiometry of motors and tubulin in individual complexes. Specifically, we did not comment on the nature of these complexes in the first manuscript. Per reviewer suggestions, we tried a number of biochemistry approaches to isolate complexes of defined number (SEC-MALS of HSET + tubulin, analytical ultracentrifugation of HSET + tubulin), but were unable to do so. We believe that these results, when considered with our fluorescence intensity results in the current iteration of the manuscript, suggest that these complexes are highly heterogeneous. Per reviewer suggestions, we have therefore updated our model to account for this heterogeneity more explicitly.

3. Concerns related to microtubule polymerization. Reviewer 3 raised a major concern regarding whether classic (GTP-driven) microtubule polymerization was required for the self-organization of microtubule asters by HSET. Of particular interest, tubulin was still able to activate HSET motors in conditions where microtubule polymerization was eliminated.

Below is a point-by-point response to reviewer comments. The reviewers' comments are in black and our responses are in dark blue.

Reviewer #1 (Remarks to the Author):

The manuscript by Norris et al. describes the surprising finding that the human kinesin-14 protein HSET can switch from a non-processive to processive mode of movement along MTs depending on the concentration of soluble tubulin in solution. This switch allows HSET to transition from a potent MT crosslinking motor that can slide anti-parallel MT bundles, a function previously well-characterized for class 14 kinesins, to one that can form stabilized MT asters *in vitro*. The proposed mechanism of this switch is the clustering of multiple kinesin motors together through interactions with soluble tubulin, similar to recently proposed motor clustering mechanisms for other kinesin-14 family members in diverse species. This discovery, further opens the broader question of what other co-factors play a role in molecular movement in cells. The results are very interesting and informative for the motor protein field, and should be

of interest to broader audiences. The experiments appear to be of high technical quality. I support publication after the authors address the following concerns:

Major Concerns:

1) The crux of the paper is the novel finding of HSET-tubulin multimers that move processively to minus-ends. The data are convincing that this phenomenon is happening in the author's experiments. However, the authors appear to gloss over the nature of these multimers in the text. They go so far as to quantify fluorescence intensities of both HSET and tubulin in these multimers, but fail to put a final number of each molecule on the moving particles (3 HSET's and 12 tubulins?). I'm guessing because of the broad range of intensities they observe?

Reviewer 1 raises an important point here. While our study defines a role for tubulin-mediated HSET clustering, we have not yet characterized specific qualities of the clusters. This includes their: 1) structural features; and 2) molecular organization, *i.e.*, the precise stoichiometry of tubulin *versus* HSET. Because of this, we did not (and could not) use strong language to describe the oligomeric nature of these HSET-tubulin complexes on our first submission. Our reasons for this are several-fold. First, Reviewer 1 is correct in that there is a broad range of fluorescence intensities of tubulin and HSET in the TIRF assay. Because of this broad intensity distribution, it is quite likely that the HSET-tubulin complexes are heterogeneous in nature. Per reviewer suggestion, we did perform SEC-MALS and AUC analysis to characterize HSET-tubulin complexes (see response to Major Point #3 below).

Because these analyses did not resolve individual peaks of higher molecular weight, it appears that HSET-tubulin complexes do not follow a prescribed stoichiometric composition. This is consistent with our fluorescence intensity results reported in **Fig. 3**. Additionally, these standard methods (SEC-MALS and AUC) may not be appropriate for detecting HSET-tubulin complexes which could occur with relatively low frequency (for further discussion, see Response to Major Point #3).

Experiments aside, we agree that the heterogeneous makeup of HSET-tubulin clusters deserves further discussion in the manuscript, and we have added further text on this point in the Discussion section.

The histograms of intensities for both HSET and tubulin show a distinct peak within the range of the single molecule intensities (Fig. 3 and S3) and the fits are not convincing for single Gaussian. The authors should provide R² values for the fits and more importantly try fitting with multiple Gaussians.

We appreciate Reviewer 1's suggestions, especially because fitting to multiple Gaussians may provide an analytical technique to uncover distinct populations within a heterogeneous distribution. For this reason, we have taken the time to address this more carefully by further analysis. Specifically, we used MATLAB to perform a least-squares fit of each intensity distribution to a multi-peaked Gaussian CDF:

$$CDF(x) = \sum_1^n \frac{w_n}{2} \left[1 + erf \left(\frac{x - \mu_n}{\sqrt{2\sigma_n^2}} \right) \right]$$

where n is the number of Gaussians, w_n is the relative weight of the n_{th} Gaussian, μ_n is the mean of the n_{th} Gaussian, and σ_n is the standard deviation of the n_{th} Gaussian. Here, we have shared this analysis for the distribution of moving Cy5-tubulin particles in the presence of GTP, corresponding to **Fig. 3d** in the manuscript, top:

As expected (each peak introduced three free parameters), the fit generally improved as the number of Gaussian peaks was increased (compare panel (a) to (b)). We visualized this by plotting the RSS (residual sum of squares) as a function of the number of Gaussian fits (c). By taking the elbow of this RSS curve, it appears that we reached a point of diminishing return around ~3 Gaussian peaks. For the distribution shown here, the means of these peaks corresponded to 66,933 a.u., 155,761 a.u., and 20,715 a.u., which roughly correspond to 7.3, 17.0, and 2.3 tubulin molecules, respectively. However, when we compare these peak locations to the intensity histogram of the distribution (d), these peaks do not qualitatively match well. We performed this analysis on all fluorescence intensity distributions in **Fig. 3**, and each intensity yielded similar results (*i.e.*, each distribution was best fit to a minimum of 3-4 Gaussians). We conclude that these fits are simply being improved by including additional free parameters, and do not appear to indicate any significant molecular structures. In other words, we could not come up with any biological justification or model for using a combination of 3 Gaussians. We find it far more likely that the distributions are simply heterogeneous, and for this reason, we have decided to eliminate Gaussian fits to the fluorescence intensity distributions altogether. Instead, we have reported the arithmetic mean and standard deviation for these distributions. We hope that this will avoid any confusion for readers and adequately address Reviewer 1's point.

How do the authors explain the processive particles that have the same intensity as single HSET molecules?

We thank the reviewer for raising this point. There is precedence for a sub-linear relationship (see, for instance, Eg5 fluorescence intensity data in Subramanian, Kapoor et al., *Cell*, 2010, Fig. 1B) between the number of fluorophores and the observed fluorescence intensity, and we believe that this applies for HSET-tubulin clusters as well. There are at least three factors which explain a systemic decrease in fluorescence intensity for these experiments, rather than truly representing single motors. First, we expect a number of fluorophores to be “dark,” which will lower fluorescence values. This is illustrated in the newly included photobleaching statistics in **Fig. S1** (response to minor point 1 – as expected, a number of the particles exhibit 1-step photobleaching). Second, we are considering the first frame of moving particles for our analysis. It is likely that HSET-tubulin complexes will occasionally bind the microtubule during the 100 ms of our single frame TIRF acquisition, thus decreasing the observed fluorescence intensity. Third, fluorophores within a cluster are likely situated closely together, and we have not accounted for any potential quenching effects which would reduce the observed intensity.

Given these factors and previous biophysical characterizations of kinesin-14 motility (Furuta, Kojima et al., 2013 *PNAS* and Jonsson, Goshima et al., 2015 *Nat Plants*), we feel confident that HSET motor activation occurs primarily via multimerization (see Discussion).

I think the molecular nature of the HSET-tubulin oligomer requires much more discussion in the manuscript.

We agree with the Reviewer, and have now expanded our discussion on the nature of these heterogeneous HSET-tubulin complexes in the revised manuscript:

“What is the molecular nature of HSET-tubulin clusters? Based on our fluorescence intensity measurements, these clusters appear to be heterogeneous in size, composition, and stoichiometry. Our AUC results suggest that either cluster heterogeneity obscures peak detection, or only a small number of clusters is sufficient to drive aster organization. Regardless, we have determined a number of key features for these clusters: (i) they contain multiple HSET motors (typically 2-6) and multiple tubulin dimers (typically 3-26), (ii) cluster formation does *not* depend on GTP-driven tubulin assembly, and (iii) clusters are likely not the result of an *in vitro* aggregation artifact. The lack of a defined cluster structure in our experiments may indicate complex formation through multivalent weak interactions resulting in condensation, as previously described for centrosomes (Zwicker, Hyman, Jülicher et al., *PNAS*, 2015). Importantly, we have shown that HSET activity in cells is modulated by the intracellular levels

of soluble tubulin, providing good evidence that our *in vitro* observations are relevant within the cell.”

2) The bigger point is can the authors provide more robust characterization of these particles to make a more convincing argument that what they are observing is a bona-fide molecular structure versus and *in vitro* artifact? Having done a lot of single molecule TIRF myself, I can say for certainty that it is very easy to find movement of aggregates within a particular prep of motor molecules which should not normally move on their own (monomers for instance). Further, tubulin is known to be sensitive to aggregation, and the addition of tubulin may introduce aggregates that nucleate motor clusters. Have the authors performed high-speed ultracentrifugation or gel filtration of their proteins (particularly tubulin) just prior to the TIRF assay as a control for aggregation-induced artifacts?

We thank Reviewer 1 for raising this important point. We have now addressed the possibility of aggregation as recommended and outlined below. Most importantly, our newly included cell-based data strongly suggests that this motor activation is not simply an *in vitro* artifact of protein aggregation (**Fig. 6**). Indeed, aggregates are always a concern for *in vitro* experiments. This concern is even greater when the protein of interest is tagged with an EGFP (or related FP), as HSET is here. In fact, this concern is very familiar to us -- the first author characterized the aggregation of FP-tagged kinesin-1 motors in a 2015 Biophys J paper (Norris, Nunez, and Verhey, 2015). Here, the possibility of aggregation may take one of three forms: either (i) HSET, (ii) tubulin, or (iii) HSET + tubulin may be prone to aggregation *in vitro*, any of which may complicate our interpretation and model. We have addressed these independently as follows:

(i) Aggregation of HSET alone. We are confident that HSET’s motility is not a consequence of an aggregation artifact of the motor itself. Each HSET construct was purified by a two-step purification (affinity followed by gel filtration). This resulted in a normal size distribution where aggregates were absent – see 1) the fluorescence intensity distribution of single HSET motors, (**Fig. 3**, n = 4840 particles), and 2) the newly included stepwise photobleaching analysis (**Fig. S1**). Second, the behavior we report here is highly dependent on tubulin. It was occasionally possible to see HSET molecules moving in the absence of tubulin – these motility events were counted and reported (see **Fig. 2E**, specifically nonzero values observed in the “No Tubulin” curve). However, these were so infrequent that we believe this behavior must be dependent on tubulin.

(ii) Aggregation of tubulin alone. For the initial submission of the manuscript, tubulin was not subjected to gel filtration or high-speed ultracentrifugation prior to use in our reconstitution assays. Rather, it was purified using standard tubulin polymerization/depolymerization cycling protocols in high-molarity buffer to remove any MAPs (Castoldi and Popov, 2003). However, to address reviewer concerns, we performed two additional experiments. First, we performed analytical ultracentrifugation on our purified tubulin by itself (Fig. S3d). Although we did detect a secondary peak at ~50 kDa (we speculate that this is monomeric tubulin), we did not detect any

significant population with a MW above ~110 kDa. We therefore concluded that our tubulin is free of aggregates. Second, in the course of performing size exclusion chromatography with multi-angle light scattering (SEC-MALS) experiments on these complexes (see below), we isolated SEC-purified tubulin as part of the “tubulin alone” control. Although our yield was quite low (we were only able to load 1.7 μM tubulin in the final experiments), we used this fraction in self-organization experiments to confirm that it could still activate HSET activity. At 1.7 μM , this gel-purified fraction was able to partially rescue HSET self-organization of GMPCPP-stabilized microtubules (similar to 2 μM tubulin as reported in **Fig. 4a, middle**):

From these experiments, we conclude that self-organization *in vitro* is *not* driven by aggregates of tubulin.

(iii) Aggregation of HSET-tubulin complexes. Finally, HSET may drive some sort of aggregation of tubulin and vice versa, which could activate its activity. However, even if this activation is driven by an aggregation mechanism, we now have evidence that this mechanism is relevant in cells (see newly included **Fig. 6**). We therefore believe that our conclusions are not simply an *in vitro* artifact. One interesting possibility is that this phenomenon may be explained by a “phase transition” or condensation mechanism at microtubule minus ends, an exciting possibility which we have now included in the Discussion.

3) The IP experiment is a decent start to show that the tail of HSET can bind tubulin, but can the authors isolate a stable HSET-tubulin co-complex by gel filtration or sucrose gradient sedimentation? It would be much more convincing if the authors could demonstrate a stable co-complex of defined molecular composition (SEC-MALS or AUC?). Or do the authors think that moving particles are heterogeneous in numbers of motors/tubulins, and that is the nature of the mechanism in cells? If so, this point should be raised in the discussion.

We thank Reviewer 1 for this comment. Notably, Reviewers 2-3 also inquired about the biochemistry/stoichiometry of these complexes. Per reviewer suggestion, we have performed both SEC-MALS and AUC and report the results here. Interestingly, we were unable to isolate defined stoichiometric co-complexes in either case. One possible explanation for this is that heterogeneous HSET-tubulin complexes would cause these peaks to broaden over a wide range of sedimentation coefficients, making them difficult to detect. This is consistent with our fluorescence intensity measurements suggesting that these complexes are heterogeneous in composition. Second, this may suggest that only a small percentage of tubulin needs to be complexed with HSET to activate its processivity, as this may be below the detection limit.

One complicating factor in these experiments limits our detection of HSET alone: HSET's extinction coefficient at 280nm is relatively low (full-length HSET contains only 2 tryptophans and 12 tyrosines). The 280 nm extinction coefficient of 6xHis-HSET is $28880 \text{ M}^{-1}\text{cm}^{-1}$ (calculated in Expasy). Invoking Beer's Law, in order to achieve a reasonable signal of ~ 0.1 AU, we would need a concentration of (assuming a 2 mm path length from the FPLC):

$$c = \frac{Abs}{\epsilon L} = \frac{0.1}{28880 * 0.2} = 17.3 \text{ } \mu\text{M} \text{ 6xHis-HSET (9.8 } \mu\text{M for EGFP-HSET)}.$$

Our typical yield from an *Sf9*-based expression and purification is on the order of 1 mL of $2 \text{ } \mu\text{M}$ for HSET, which is below this threshold. Because both our SEC-MALS FPLC and our AUC are based on 260/280 nm detection, and both approaches require ~ 0.5 mL sample, we were unable to detect HSET alone. Therefore, we instead relied on the signal from tubulin in the hopes that its peak location would shift after the addition of HSET.

First, we attempted a SEC-MALS approach to detect these complexes. We injected a sample of $9.5 \text{ } \mu\text{M}$ tubulin alone (red curve), $1.4 \text{ } \mu\text{M}$ EGFP-HSET alone (green curve), and $9.5 \text{ } \mu\text{M}$ tubulin + $1.4 \text{ } \mu\text{M}$ EGFP-HSET (orange curve), and monitored their elution by A280:

As expected (see above), HSET alone was undetectable at this concentration. Tubulin showed a prominent peak at ~100 kDa, consistent with that of a dimer. Interestingly, tubulin (and tubulin + HSET) also showed a peak that eluted in the void volume. We suspect that this is due to microtubule assembly, and not some non-specific aggregation, because: (i) the MALS laser must be used at room-temperature, so these samples could not be kept on ice during the elution, (ii) there is likely some contaminating GTP in our tubulin prep (see response to Reviewer 3 for a further discussion on nucleotide state), and (iii) analytical ultracentrifugation of tubulin alone at 4°C, which is a far more sensitive approach, showed essentially no population with a MW > 110 kDa (see next paragraph). Regardless, the tubulin + HSET sample was indistinguishable from tubulin alone and showed no additional peaks. From this experiment, we concluded that either: (i) tubulin-HSET complexes are sufficiently heterogeneous that peaks were not resolvable by SEC, (ii) these complexes were so infrequent compared to dimeric tubulin that they were not detectable by SEC, or (iii) some combination of these two factors. In an attempt to address this with higher resolution and detection, we moved forward with a similar experiment using analytical ultracentrifugation.

Because HSET was undetectable by absorbance at 280 nm (see above), our approach was to ask whether the addition of 200 nM EGFP-HSET would shift the sedimentation profile of 5 μM tubulin. We were unable to detect a signal above background at higher sedimentation coefficients than the values for the tubulin dimer, and this result has been included as Fig. S3e:

We thus concluded that, consistent with the SEC experiment, HSET and tubulin do not form a complex of a prescribed stoichiometry. Rather, although these proteins interact strongly at interactions much lower than these experiments (see Fig. 3a), they must be arranged in a heterogeneous composition that is not strongly detectable by these techniques. Along these lines, the motile fraction on GMPCPP MTs (see Fig. 2) likely represents a relatively small fraction of the overall tubulin in solution.

4) I think Fig. 3 could be revamped. I don't find the images of particles in B and C very enlightening. I would replace the particle images with the data from Fig. S3B, C.

We thank the reviewer for their suggestion and have incorporated their feedback. We do believe that the particle images are still valuable for the readers to assess our approach, so we have moved them to **Supp. Fig. S3B-C**.

I'm also confused how they got the brightness measurements at 10nM Cy5-tubulin. Is 10nM the total tubulin concentration in this experiment, which seems oddly low since all the other processive movement is seen at much higher tubulin concentrations? I have the same question for Fig. 2G and Fig. S3C.

Yes, 10 nM is the total tubulin concentration in these experiments. In general, we adjusted the amounts of tubulin and HSET as necessary depending on the objective of each experiment. For these experiments, our goal was to resolve individual particles of tubulin so it was kept to a minimum. In other experiments where the goal was to examine HSET activation, we studied motor processivity with much higher (μM) levels of tubulin. Additionally, the tubulin concentration for these Cy5-tubulin experiments is lower than experiments where EGFP-HSET was used because we used unlabeled HSET (no N-terminal EGFP tag) for these experiments. It appears that the unlabeled HSET motors were activated by tubulin more efficiently than equivalent levels of EGFP- HSET, so we reason that the N-terminal EGFP tag might partially inhibit tubulin binding. Finally, we also noticed that the concentrations of tubulin/HSET were not explicitly mentioned in every figure sub-legend, so we have updated this accordingly.

The efficiency of tubulin labeling should be reported as well, as this is critical for the calculation of how many tubulins are in a given spot. Have the authors taken this into account?

We labeled tubulin in its polymeric form (and subsequent cycling) using NHS esterification and used 100% labeled tubulin (1 dye molecule per dimer) in our assays. Because there is no unlabeled tubulin present, we do not need to account for “dark” tubulin in our analysis. Even if there is some variability in how each molecule is labeled (i.e., some dimers have 2 dyes and some have 0 dyes), this statistical variation should happen at the same rate for each dimer, and therefore we have not considered this in our fluorescence intensity calculations.

5) The self-organization assay is convincing and a nice addition that reconciles past observations. How do the authors envision that HSET “toggles” between modes of motility in vivo given the high intracellular concentration of soluble tubulin? A sentence or two on this point seems warranted.

We believe that this is an important point that is central to HSET's function. In addition to addressing this in the Discussion, we also include new data, showing that acute accumulation of unpolymerized tubulin in cells causes HSET to drive aster formation. For this experiment, we generated a cell line in which EGFP-HSET could be induced through simple addition of doxycycline to the media. We estimate (by immunoblot analysis) that EGFP-HSET levels peak

at ~4-fold above endogenous HSET protein levels. Our idea for this experiment was as follows. If soluble tubulin causes HSET to organize microtubules into asters, then acute polymer disassembly via nocodazole might convert bipolar spindles to acentrosomal asters. This would indicate that the motor was able to toggle between its modes of motility, and this was indeed the outcome of the experiment (**Fig. 6**).

Certainly, it is true that cells contain high concentrations of soluble tubulin at all times. An important point to consider is that HSET is not the only factor which is likely to be sensitive to the concentration of soluble tubulin. In addition, we also do not know the rates at which soluble and polymeric tubulin interconvert. It seems unlikely that HSET is exclusively limited to only one mode of action at a time (i.e., filament sliding versus processive motion), and it instead exists in an equilibrium between the two states. Along these lines, we are currently envisioning a model similar to that of the “actin economy” in cells, where F-actin and G-actin differentially activate cytoskeletal factors (Papakonstanti, Vardaki, and Stournaras, *Cell Physiol Biochem*, 2000). We have included these points in the Discussion, and have updated our model figure (**Fig. 7**) to make this point more clear.

Minor Points:

1.) Fig. S1. Two-step photobleaching is a good start towards demonstrating the oligomeric state of the HSET preparation. However no statistics are given about the data. What fraction of motor spots showed two-step vs. other steps of bleaching, and how many spots were counted/excluded. In my experience, step photobleaching can be tricky and not very robust. It would be good if the authors had another confirmation of their protein state (intensity analysis compared to another dimeric, GFP-tagged protein?).

We thank the reviewer for their suggestion. We have included these statistics for all constructs in the paper ($n > 200$ events for each construct), and compared it to EGFP-XMCAK, a known dimer.

2.) Fig. 3B, C. I was surprised that the authors do not put a final estimated number of molecules on their quantifications of motor and tubulin intensities, even after starting the text section by stating that they counted numbers of molecules.

See Major Point #1. We have included more specific language in the text.

3.) Although importin α/β is a likely regulator of HSET via binding to its tail domain (38), this mechanism would more likely act as a global on/off switch for HSET activity rather than toggling between MT sliding and processive motion. The authors should expand on this point as it seems very relevant to their mechanism.

We have included an expanded discussion on HSET's toggling activity in the Discussion (see response to Major Point #5).

4.) Fig. 2H is mislabeled

We apologize for the oversight and have fixed the panel letter.

5.) The authors claim to visualize HSET and tubulin "simultaneously" in Fig. 2G and Movie 5. Is this true simultaneous imaging? If so, the authors have failed to describe the microscope setup for imaging two channels simultaneously (i.e. are they using an image splitting device or two cameras).

We apologize for the oversight and have updated our methods section to say "near-simultaneous" rather than "simultaneous." We are using triggered acquisition via AOTF to switch between 488 nm and 561 nm excitation, and these channels are offset by 100 ms.

6.) The authors claim that there are approximately three HSET-FL molecules per Qdot in their experiments, but the numbers in Fig. S4 look more like two motors per Qdot.

We apologize for the oversimplification. On average, the FL-HSET QDots contain ~2.3 motors, and the tailless HSET QDots contain ~3.4 motors. We have added a statement clarifying this in the text.

7.) The tubulin channel in movie S4 seems strangely cropped compared to the HSET channel in the 1nM concentration panel.

We have chosen a different microtubule from the same condition that is not near the edge of the image, and have updated movie S4 accordingly. We have also increased the temporal resolution and eliminated Fiji's .avi compression, both of which allow the movie to play more smoothly.

8.) The authors make the point to mention that the fraction of moving motors increased as concentration of HSET increased. Do they suggest cooperativity or interactions between HSET molecules?

We do believe that this interaction is cooperative, and would like to point out that Reviewer 3 raised this point as well. Our feeling is that we are sampling only the "bottom elbow" of cooperative interaction. For a truly accurate measurement of cooperativity, we would need to explore the effect at higher HSET concentrations. However, individual motile events are almost

impossible to resolve at this concentration and above (see **Fig. S2b**). It is worth noting that, for this reason, the event frequency at 20 nM is likely an underestimate. We have eliminated the connecting line in **Fig. 2e** to avoid misleading readers.

9.) The authors report dwell time data at minus-ends with Qdots only. It is not clear what the point of this data is in the current manuscript. I would recommend performing dwell time analysis for native clusters. If the numbers are similar, it is an additional parameter to show Qdots are effectively mimicking the tubulin induced clustering of motors.

We believe that HSET's end-dwelling capability is essential for its self-organization of higher-order structures, and we apologize that this was not clear in the initial manuscript. Our reasoning is that tailless HSET conjugated to QDots does not drive self-organization (**Fig. 5e**), and its end-dwelling capability is significantly reduced relative to full-length (FL) HSET. In terms of measurable parameters, the ability to dwell on MT ends seems to be the most tangible difference between FL HSET and tailless HSET when conjugated to QDots. Further, it is expected that the ability of the motor to *accumulate* at MT minus ends would help to drive higher-order organization by providing a template for cargo transport and MT focusing (Nedelec et al., *Nature*, 1997).

Originally, we did not include the end-dwell analysis for native clusters because of lower n values relative to the QDots (i.e., number of particles that arrived at MT minus ends and were analyzable during the time of our observation). Essentially, even with hundreds of processive runs, the only data points available to analyze are the light-red events from **Fig. 2f**. Upon further consideration and at the reviewer's request, we believe this end-dwell analysis is important to the story. We revisited this data set and have plotted the end-dwell times for the native clusters, now included as **Fig. S2f**. Indeed, the end-dwell distribution is similar (25 ± 8 s) to that of the full-length motor on QDots (37 ± 16 s). We have also added a sentence in the manuscript addressing this discussion point.

Reviewer #2 (Remarks to the Author):

In this work, the authors study the formation of microtubule asters by the human kinesin-14, HSET. They found that HSET can form asters from dynamic microtubules, but cannot form asters from stabilized (i.e. non-dynamic) microtubules. They explain this finding by the fact that HSET, a non-processive motor, becomes processive in the presence of soluble tubulin through the formation of multi-motor tubulin-HSET clusters.

We thank Reviewer 2 for his/her feedback and have attempted to address these major concerns below.

The transition from the non-processive state to the processive might be an interesting switch mechanism, the authors however at the moment fail to present convincing evidence that the clustering, which drives the switch, is not a result of rather non-specific protein aggregation. Any aggregates of a non-processive motor are likely to become processive.

Reviewer 1 had similar concerns which were addressed in detail above. In summary, through additional experiments, we can now confidently say that this switch is *not* a result of either (i) aggregation of HSET alone or (ii) aggregation of tubulin alone.

To rule this out, the authors should investigate the clusters further, for example by i) showing in detail how are these clusters assembled (e.g. by biochemical mapping of the interaction interfaces going beyond the current identification of the HSET tail-domain), ii) investigating the structure of the clusters (e.g. by electron microscopy), and/or iii) showing that similar clustering occurs in the cell.

We appreciate Reviewer 2's suggestions and have attempted a number of experiments to address these concerns (addressed in reverse order here).

(iii) Showing that similar clustering occurs in the cell. We are pleased to report that we tested our model in cells and we observe a similar phenotype to our self-organization experiments *in vitro*. In short, we used low doses of nocodazole to partially depolymerize microtubules, which rapidly increases the amount of soluble tubulin relative to polymer. As predicted by our model, we observed microtubule aster formation under these conditions within ~15 min. This effect was specific to HSET, as the effect only occurred on these timescales when EGFP-HSET was overexpressed at ~4-fold relative to wild-type. These experiments are now included as **Fig. 6** in the revised manuscript, and we hope that this alleviates any concerns that HSET motor activation was an *in vitro* artifact.

(ii) Investigating the structure of the clusters (e.g., by electron microscopy). Additionally, we performed two standard biochemistry approaches (SEC-MALS, analytical ultracentrifugation) in an attempt to isolate pure HSET-tubulin complexes of defined number (see responses to Reviewer 1). Based on these experiments (in addition to the quantitative fluorescence approaches in the manuscript), it appears that these HSET-tubulin complexes are too heterogeneous to isolate a specific population. We were therefore unable to perform any structural biology (e.g., electron microscopy) on these complexes in the time frame allotted for revisions. We do, in fact, believe that a structural approach is important for understanding HSET motor activation; to this end, we are actively pursuing these approaches for future studies.

(i) Showing in detail how are these clusters assembled (e.g. by biochemical mapping of the interaction interfaces going beyond the current identification of the HSET tail-domain). The interaction between the HSET tail domain and tubulin is at least partially electrostatic in nature, given tubulin's well-known negatively-charged surface, and the net positive charge of HSET's N-terminal tail domain (net charge of +17). Along these lines, at high salt concentrations, the HSET tail domain showed diminished binding to GMPCPP-MTs in TIRF assays (see response to

Specific Comment #2 below). However, the interaction between the HSET tail and soluble tubulin still occurs with ~nanomolar affinity even at very high ionic strength (*e.g.*, 80 mM PIPES + 50 mM KCl at 4°C, see **Fig. 3a**). We therefore believe that, although charge-based, the interaction between HSET's tail and tubulin must occur with a reasonable degree of specificity. Our opinion is that mapping of the specific binding site is beyond the scope of this current work within the allotted time frame for revisions, and we will be pursuing these studies in the future.

Moreover, aster formation of dynamic microtubules by kinesin-14 has already been shown by Hentrich and Surrey for XCTK-2 (ref. 1 in the manuscript). The additional insight provided by the current manuscript is rather marginal, given that no mechanistic understanding of the cluster formation (beyond the purple ellipses in Fig. 5B) is offered. I can therefore not recommend consideration for publication in Nature Communications.

Indeed, aster formation of dynamic microtubules by kinesin-14 was previously shown by the Surrey lab. Interestingly, in an older study (Nedelec, Surrey, Maggs, and Leibler, 1997, *Nature*), the authors showed that artificially-formed clusters of *plus*-end directed motors (formed by mixing biotinylated kinesin-1 and streptavidin, thus presumably groups of ~4 motors) were able to self-organize pre-formed microtubules into asters so that their plus-ends were directed inwards. In subsequent studies, the authors looked at the interplay between plus- and minus-end motors (Surrey and Karsenti, 2001, *Science*), but again, these motors were artificially clustered (this time by antibodies). Thus, the previous work left open the question of how motor clustering may occur in a physiological context.

Importantly, in the Hentrich and Surrey paper (ref. 1) using *Xenopus* kinesin-14, no artificial clustering was used, but these experiments were performed using *dynamic* microtubules where free tubulin would have been present in excess. However, this particular aspect of the study has not been discussed. We therefore believe that our current manuscript addresses a *critical* knowledge gap in the field: the kinesin-14-tubulin clustering mechanism described here is the *reason* why no artificial clustering is necessary for self-organization in these dynamic microtubule experiments. Perhaps most importantly, based on reviewer suggestions, we have now addressed this phenomenon in cells, which has not been attempted in previous studies. Including this new data has necessitated substantial text edits throughout the Discussion to make this important point more clear to readers.

Specific comments:

1) The authors say that "These particles are unlikely to be aggregates, because: (i) moving particles appeared as diffraction-limited spots... (iii) the intensity distributions were still well-fit to a normal distribution..."

Neither seems to be the case, see figs. 3B,C

We appreciate Reviewer's comment and apologize for the particularly poor choice to use a normal distribution to fit our fluorescence intensity data. We did try fitting sums of Gaussian distributions as well (see response to Reviewer 1, Major Concern #1), but were ultimately unable to come up with any scientific justification for doing so. We have thus chosen to eliminate any fitting to these data and instead we have reported arithmetic means and standard deviations.

We do find that both the tubulin and HSET particles move predominantly as diffraction-limited spots in our TIRF experiments. However, Reviewer 2 (and Reviewer 3, see below) raises an important point here: even if these particles *were* aggregates of some sort, they would still appear as diffraction-limited if they have a small-enough diameter. Therefore, we agree that this is not the best justification for ruling out protein aggregation, and we have eliminated this point from the text.

Rather, we have now performed additional experiments to investigate potential aggregation - please see the response to Reviewer 1 (Major Concern #2). Based on these additional experiments, we strongly believe that we can rule out protein aggregation as an artifact that drives the clustering behavior.

2) The authors never comment on the structure of the clusters. Does the HSET tail bind to several tubulin dimers? If so, why wouldn't the clusters grow further with increasing concentration of HSET/tubulin? In fact, it would not surprise me, if a similar cluster generation and microtubule aster formation would also occur in the presence of other negatively charged proteins (e.g. free globular actin instead of tubulin).

Please see the responses to Reviewer 1 as to why we did not comment on the structure of the clusters. As of the first submission, we did not feel comfortable speculating on the specific stoichiometry of the interaction. After additional experiments as suggested by all reviewers, we are confident that the HSET-tubulin clusters are heterogeneous in size and composition. We have made this point more clear in the revised Discussion. We have also updated the model figure to reflect the heterogeneous nature of these clusters.

To address Reviewer 2's second point, we believe that the casein in our assay provides an extremely important internal control for this proposed mechanism. All *in vitro* experiments were performed in 0.5 mg/mL casein in pH \geq 6.8 buffer. Casein is even more acidic than tubulin – the isoelectric point of casein is between 4.1 and 5.8 (Sigma C5890 product sheet), and the isoelectric point of tubulin is between 5.2 and 5.8 (Williams, Shah, and Sackett, 1999, *Analytical Biochemistry*). Therefore, all experiments are performed in an excess of negatively charged protein at all times. Because we only observed cluster generation/aster formation in the presence of soluble tubulin (and never in the absence), we conclude that the activation of HSET is specific to tubulin.

3) The ionic strength of the measurement buffer is known to affect protein aggregation. Why

- were the experiments performed in three different buffers / three different ionic strengths?
- self-organization in 20 mM K-PIPES, 50 mM KCl
 - single-molecule diffusion experiments 12 mM PIPES
 - all other experiments studying motility behavior BRB80 buffer containing 50 mM KCl

We apologize if this was confusing to readers and we appreciate Reviewer 2's attention to detail. Our "default" buffer for any *in vitro* experiments is the third buffer listed here (BRB80 + 50 mM KCl, see e.g. Zanic et al. NCB 2013, Zanic Methods Mol Bio 2016). We typically use this buffer due to its relatively high ionic strength, which helps to dampen any nonspecific ionic interactions. Any charge-based protein aggregation is thus less likely in this buffer, and we have used this buffer whenever possible unless otherwise noted. The exceptions are as follows:

1) Single-molecule diffusion experiments, 12 mM PIPES. We used this buffer here for two reasons. First, it had been previously reported that Ncd (fly kinesin-14) was very weakly processive at very low ionic strengths (Furuta and Toyoshima, 2008, *Current Biology*). We wanted to carefully examine any potential processivity of the motor alone, and thus opted to perform the single-molecule experiments in a buffer with comparably low ionic strength. Second, the binding affinity of the HSET Δ Motor construct was reduced quite a bit in our default buffer, and obtaining MSD measurements was impossible under these conditions. In order to make the claim that this diffusion was driven by the tail domain, performing our measurements in the 12 mM PIPES buffer was necessary here. We have now included a sentence citing this 2008 paper in the text to justify our buffer selection.

2) Self-organization experiments, 20 mM PIPES + 50 mM KCl. This is the field-standard "self-organization" buffer (20 mM PIPES + 50 mM KCl) that was previously used in established self-organization assays (ref). To best replicate conditions previously reported in the literature, we decided to also use this buffer in our self-organization assay.

Significantly, in the newly included **Fig. 6**, we have now demonstrated that soluble tubulin plays a role in the activation of HSET in cells, thereby minimizing concerns related to buffer composition *in vitro*.

4) Minor: Figure "2H" should probably be "2G".

We apologize for the oversight and have corrected the error.

Reviewer #3 (Remarks to the Author):

In this manuscript the authors investigate the role of the kinesin14, HSET, in forming microtubule asters *in vitro*. The data nicely show that in single molecule assays, HSET is non-

processive; that adding tubulin to HSET increases processivity; that tubulin co-localizes with processive motors; that adding tubulin to HSET and stable microtubules can drive aster formation; and that HSET on a Q-dot mixed with stable microtubules also results in aster formation.

We thank Reviewer 3 for his/her feedback. We have addressed each concern individually below.

My main concern is the nature of the tubulin that interacts with the motors and the stoichiometry of the motor tubulin complex. In the experiment in which Qdots were used there is no soluble tubulin: the motors are linked to the Q-dots via the streptavidin-biotin-his tag on the motor and added to stabilized microtubules. Under these conditions, asters form. This result suggests that HSET tail can bind to a cargo microtubule because tailless HSET on the Q-dot fails to drive asters from stable microtubules. However, other experiments show that GTP is not needed for inducing aster formation from a mixture of motors, stable microtubules and tubulin, suggesting that it is soluble dimers, not microtubules, that are needed to bind to the motor, induce clustering. Is the GTP totally absent from the solution in this experiment? Do the authors envision that one dimer binds to two (or more) motors, generating a cluster? It would seem more likely that the soluble tubulin forms a short oligomer or protofilament of some sort, which could then bind to more than one motor, generating a cluster. One control experiment comes to mind: adding colchicine to the tubulin will prevent oligomerization. Does tubulin-colchicine, HSET and stable microtubules result in aster formation? In many places in the manuscript the authors state that HSET tail binds soluble tubulin, but is soluble tubulin responsible for cluster formation?

Reviewer 3 raises an excellent series of questions essential to the model of HSET-tubulin activation we have proposed, and we have performed some additional experiments based on this reviewer's request. Based on these experiments, we strongly believe that soluble tubulin, rather than MTs polymerized via a canonical GTP-driven mechanism, is the agent that drives HSET clustering and activates the motor for self-organization.

In short, we favor a model (see updated model **Fig. 7**) where HSET can interact with both soluble tubulin (via its tail domain only) *and* microtubule polymers (via either its tail domain or its motor domain). The relative availability of each of these substrates would, in theory, dictate the activity of the motor, and the motor exists in an equilibrium between the two states. Our data indicate that either (i) artificial oligomerization by conjugation to a QDot (full-length or tailless motor), or (ii) the addition of soluble tubulin to the (full-length) motor drive motor clustering and activation of processivity. However, as Reviewer 3 has noted, the HSET Δ Tail conjugated to QDots does not promote aster formation. The simplest explanation for this result is that the ability of the full-length motor to dwell at MT minus-ends plays a large role in this (see response to Reviewer 1, Minor Point #9). Another contributing factor (raised by the Reviewer 3 here) is that the tail is seemingly still able bind cargo MTs after the motor has been clustered. In other words, the tail domain of HSET clusters may have enough tubulin/MT binding sites to bind both soluble tubulin *and* MTs simultaneously. This is plausible, but requires additional work to determine how the binding interface between the HSET tail and tubulin is arranged. For

example, even though there are, on average, ~4 tubulin dimers present per dimeric motor, it is possible that the HSET tail still has an unoccupied binding site allowing it to bind MT polymer as a cargo. Unfortunately, we cannot currently think of a way to test this without a significant step forward (see response to Reviewer 2). We have therefore opted to include a few brief sentences on this in the revised Discussion.

Reviewer 3 raises an important point about a potential requirement for GTP for HSET self-organization. To investigate this further, we performed two experiments that were inspired by the reviewer that we have included in the manuscript. We performed self-organization experiments using GMPCPP MTs under the same conditions as before (see Fig. S4), but we made a number of changes. First, we (i) eliminated the 2 μ M taxol, (ii) replaced 1.5 mM GTP with 1.5 mM GDP, and (iii) added 100 μ M colchicine. Second, we (i) eliminated the 2 μ M taxol, (ii) replaced 1.5 mM GTP with 1.5 mM GDP, and (iii) added 33 μ M nocodazole. In both cases, we expect no MT polymerization to be possible, as there are saturating amounts of both non-hydrolyzable guanine nucleotide (GDP) and tubulin-sequestering compound (colchicine or nocodazole). Interestingly, in both cases, self-organization by HSET proceeded rapidly (~minutes) after the addition of tubulin. This provides strong evidence that classic GTP-driven polymerization is not required for the activation of the motor by tubulin. Note that this is also consistent with our result that omitting GTP still led to the transport of tubulin clusters in our TIRF assay (**Fig. 3d**). We therefore conclude that soluble tubulin *drives* cluster formation. We appreciate the suggestion and have included these experiments as **Fig. 4b** and **Fig. S4b-c**.

Related to this concern about tubulin vs a microtubule functioning to form clusters, is the data in Figure 3. The distribution of fluorescence for the motile motors and tubulin shows a very wide range of values. It is not clear if there is some preferred stoichiometry, nor is it clear if some of these are aggregates (even multiple molecules, of say 8nm each, can be diffraction limited spot). And if there are indeed ‘aggregates’ or oligomers, maybe this is required to induce clustering. This experiment was done with a relatively low concentration of HSET; what happens at higher concentrations of motor, or lower concentrations of tubulin – does the distribution of fluorescence values shift?

We thank Reviewer 3 for his/her concern. Please see the response to Reviewer 1 (Major Concern #2) for a lengthy discussion of protein aggregation.

We did try a number of different concentrations of HSET and tubulin when quantifying our fluorescence values (see newly modified **Fig. 3**). Combinations of (1 nM EGFP-HSET + 20 μ M tubulin) and (10 nM EGFP-HSET + 2 μ M tubulin) yielded nearly indistinguishable intensity distributions. Because this is consistent over very different stoichiometric ratios of tubulin and HSET (20,000:1 and 200:1, respectively), we are confident that this interaction is not an artifact of aggregation. It is worth noting that we have also tried quantifying the fluorescence intensity using even higher levels of EGFP-HSET (not included), but we could no longer distinguish individual particles and thus were unable to obtain individual intensities.

The authors need to show if soluble tubulin in a solution that has no GTP or that has colchicine, is sufficient for aster formation or if the tubulin is forming a higher order structure which then clusters the motors.

We thank Reviewer 3 and agree that this is an important point. We performed this experiment (as well as an analogous experiment with nocodazole) and were able to show that soluble tubulin still drives aster formation. See the previous response point for a longer discussion of this. This result is now included as **Fig. 4b, Fig. S4b-c**.

Other concerns:

The data in Figure 2 show that processivity is enhanced with higher concentrations of HSET, but only 10nM and 20nM are shown; what happens at 15nM? It seems important to include another concentration to learn if there is a linear relationship for this experiment or not.

We thank Reviewer 3 for their concern. The precise nature of the relationship (linear *versus* non-linear) is less important than the point that the number of motile events increases as a function of motor concentration. We did not intend to mislead the reader regarding the conclusion from this experiment, and have therefore eliminated the dotted line between data points in **Fig. 2e**.

When discussing adding a Q-dot to HSET, the text does not mention that the Qdot is a Streptavidin tagged etc. The text should include this information.

We initially included this information in the Materials and Methods and the schematic (**Fig. 5a**), but neglected to mention it in the text. We apologize for the oversight and now mention it specifically.

Figure legend does not mention panel H.

We apologize for the mistake and have updated it accordingly.

Figure 3 has no molecular weight markers on the blot.

We apologize for the oversight and have noted the molecular weights on the blot (as well as the newly included blot in **Fig. 6a**)

The manuscript refers to asters and bundles of microtubules as “structural motifs”. I do not think that this is an accurate use of that term. Could the authors please consider describing the micron

scale organization of microtubules in a different way, for examples, ‘microtubules can be organized into different architectures’, or into ‘different micron scaled structures’, or simply that microtubules can be organized into bundles and asters.

Reviewer 3 raises an interesting point here. We agree that “structural motifs” sounds more like we are describing alpha helices or beta sheets. We have taken the reviewer’s suggestion of “architectures” and implemented it in the text.

I like the term ‘cluster’ not ‘team’; the latter sounds like draught animals or a sports metaphor. Please switch to cluster.

We appreciate the suggestion and have updated the text accordingly.

Reviewers' comments:

Reviewer #1 (Remarks to the Author):

The authors have done a nice job responding to my initial comments. The nature of the tubulin:HSET oligomers clearly requires more work, but this manuscript opens that door. The addition of the in vivo data is interesting and supportive of the in vitro work. I support publication of the revised manuscript and think the work will stimulate much interest and further research in the field.

Reviewer #2 (Remarks to the Author):

The current manuscript is a greatly revised version of the original submission. The authors have added numerous new in vitro data in response to the reviewers' comments. Moreover, in vivo data has been added, showing that after nocodazol treatment, which partly depolymerizes microtubules, new microtubule asters are formed. The authors show that the aster formation is dependent on the HSET amount in the cytoplasm (WT vs HSET over-expression). As such, the described phenomenology is now very clear.

Nevertheless, a significant number of questions with regard to interpreting the data and hypothesizing a plausible model still persist in my eyes. While I understand that a full structural characterization of the motor-tubulin clusters may be beyond the scope of the current work, I do encourage the authors to go a bit further in developing (and potentially testing) ideas about the underlying principles.

I do regard the manuscript potentially suitable for publication after addressing the following comments:

Major comments:

1) At numerous places (e.g. in lines 95, 123, 263, 264, 270, 273... but also at other locations) the authors state that "HSET motor processivity is activated by soluble tubulin" (line 95) or that "HSET may exist in equilibrium between two configurations" (line 123) or that "soluble tubulin induces HSET processivity" (line 263). I find this quite misleading as it suggests that soluble tubulin would change the properties of the individual motors. However, this is not the case. Rather, it should be consistently stated that "the presence of soluble tubulin causes the clustering of HSET and that these clusters show processive motility (as compared to the non-processivity of single HSET motors)".

2) The authors rule out that aggregation of either tubulin or HSET alone is present and necessary for the formation of the tubulin-HSET clusters. While indeed a number of observations hint to the absence of pure tubulin or HSET aggregates, how do the authors envision the clusters are forming? The pure interaction of one HSET dimer with (exactly) one tubulin dimer would not lead to cluster formation. Rather, it would be necessary that one tubulin dimer interacts with at least two tail domains from two different motors ... or in other words, each HSET dimer needs to bind to more than one tubulin dimer. If the authors believe this is the case, it should be straightforward to test this hypothesis. For example: immobilize full-length GFP-HSET in AMPPNP to surface-immobilized microtubules (which are either (i) not labeled and detected by DIC/darkfield or (ii) labeled by a third color) and quantitatively image the binding of fluorescently-labeled tubulin to these GFP-HSET molecules.

This experiment would answer another open and important question that should be addressed, namely if the binding of tubulin to HSET-tail is transient or permanent.

Along the same lines, if neither tubulin nor HSET aggregates were involved, it should be possible to image the dynamics of cluster growth (starting from single molecules of GFP-HSET and/or Cy5-tubulin all the way up to clusters of limited final size) in TIRF.

What do the authors think limits the size of the clusters?

If the behavior described cannot be shown, the only alternative hypothesis I can think of is that tubulin is partly present in an aggregated form (actually still my favorite hypothesis). Even if only a small fraction of the total tubulin is aggregated, these aggregates (after collecting multiple HSET motors) will be fished out of solution by their prolonged interaction with the surface-immobilized microtubules. In a way, that is what the authors somehow show in their model in Figure 7c. Moreover, I believe the presence of such aggregates (or microtubule fragments) is not unlikely, both in vitro and in vivo.

The speculation that "The lack of a defined cluster structure in our experiments may indicate complex formation through multivalent weak interactions resulting in protein condensation, as previously described in multiple intracellular contexts" is quite far-fetched, as the experimental conditions for observing such condensates in vitro are quite different from the present ones.

3) It has been shown before that microtubule destabilizing drugs, such as nocodazol, can induce aster formation (e.g. Visualization of Aberrant Perinuclear Microtubule Aster Organization by Microtubule-Destabilizing Agents, Shinji SAKAUSHI, Kaori SENDA-MURATA, Shigenori OKA & Kenji SUGIMOTO, *Bioscience, Biotechnology, and Biochemistry* 73(5), 1192-1196, 2009). This paper should be cited and discussed.

Further comments:

4) Line 20: "growing MTs" ... I guess it is more important emphasize the presence of free tubulin

5) Lines 106-108: "and that tail deletion is insufficient to form a constitutively active motor." I believe this statement refers to potential auto-inhibition (as explained later). This may, however, not be clear to the reader at this point.

6) Lines 118-125: I did not get the idea why the authors wanted to visualize single HSET molecules here. Why would that be necessary as compared to imaging the clusters?

7) Line 129: replace "far-red" by "Cy5"

8) Figure 7b: Why would HSET bound to the microtubule via its motor domain diffuse on the microtubule (small arrows)?

9) A recent paper in *Nature Chemical Biology* (doi:10.1038/nchembio.2495) reports on the activity of HSET during microtubule-microtubule sliding (along with a single-molecule characterization of HSET, see Fig. 3). Although there is no direct overlap to the scope of the current manuscript, the authors may want to consider discussing that paper (for example with regard to the finding that HSET appears to interact slightly different with microtubules as compared to other kinesin-14s, such as Ncd).

We thank the reviewers for their additional comments and suggestions. Below is a point-by-point response. The reviewers' comments are in black and our responses are in dark blue.

Reviewer #1 (Remarks to the Author):

The authors have done a nice job responding to my initial comments. The nature of the tubulin:HSET oligomers clearly requires more work, but this manuscript opens that door. The addition of the in vivo data is interesting and supportive of the in vitro work. I support publication of the revised manuscript and think the work will stimulate much interest and further research in the field.

We thank reviewer 1 for their response. We agree that the oligomeric nature of the HSET-tubulin clusters is deserving of further study and will investigate this in the future.

Reviewer #2 (Remarks to the Author):

The current manuscript is a greatly revised version of the original submission. The authors have added numerous new in vitro data in response to the reviewers' comments. Moreover, in vivo data has been added, showing that after nocodazol treatment, which partly depolymerizes microtubules, new microtubule asters are formed. The authors show that the aster formation is dependent on the HSET amount in the cytoplasm (WT vs HSET over-expression). As such, the described phenomenology is now very clear.

Nevertheless, a significant number of questions with regard to interpreting the data and hypothesizing a plausible model still persist in my eyes. While I understand that a full structural characterization of the motor-tubulin clusters may be beyond the scope of the current work, I do encourage the authors to go a bit further in developing (and potentially testing) ideas about the underlying principles.

I do regard the manuscript potentially suitable for publication after addressing the following comments:

Major comments:

1) At numerous places (e.g. in lines 95, 123, 263, 264, 270, 273... but also at other locations) the authors state that "HSET motor processivity is activated by soluble tubulin" (line 95) or that "HSET may exist in equilibrium between two configurations" (line 123) or that "soluble tubulin induces HSET processivity" (line 263). I find this quite misleading as it suggests that soluble tubulin would change the properties of the individual motors. However, this is not the case. Rather, it should be consistently stated that "the presence of soluble tubulin causes the clustering

of HSET and that these clusters show processive motility (as compared to the non-processivity of single HSET motors)”.

Reviewer 2 raises an excellent point, and we understand that this language could be misleading. Our intention was to describe the motility of HSET ensembles, rather than to suggest a mechanism for the activation of individual motors within clusters. Where appropriate, we have modified the text to rephrase this using the suggested language, and we thank the reviewer for their suggestion.

2) The authors rule out that aggregation of either tubulin or HSET alone is present and necessary for the formation of the tubulin-HSET clusters. While indeed a number of observations hint to the absence of pure tubulin or HSET aggregates, how do the authors envision the clusters are forming? The pure interaction of one HSET dimer with (exactly) one tubulin dimer would not lead to cluster formation. Rather, it would be necessary that one tubulin dimer interacts with at least two tail domains from two different motors ... or in other words, each HSET dimer needs to bind to more than one tubulin dimer. If the authors believe this is the case, it should be straightforward to test this hypothesis. For example: immobilize full-length GFP-HSET in AMPPNP to surface-immobilized microtubules (which are either (i) not labeled and detected by DIC/darkfield or (ii) labeled by a third color) and quantitatively image the binding of fluorescently-labeled tubulin to these GFP-HSET molecules.

This experiment would answer another open and important question that should be addressed, namely if the binding of tubulin to HSET-tail is transient or permanent.

Along the same lines, if neither tubulin nor HSET aggregates were involved, it should be possible to image the dynamics of cluster growth (starting from single molecules of GFP-HSET and/or Cy5-tubulin all the way up to clusters of limited final size) in TIRF.

We thank the reviewer for their comments and have performed the suggested experiment, described in further detail below. Because of the relative stoichiometry of HSET and tubulin dimers in clusters (~3:14), as quantified by our fluorescence measurements, it is possible that a single HSET motor directly binds only a single tubulin dimer, while the tubulin dimers may be interconnected. Thus we do not necessarily expect each motor in the cluster to be bound to multiple tubulin dimers. Broadly speaking, we do hypothesize that cluster formation is occurring in solution, rather than on the polymeric microtubule surface. Our primary evidence for this is that the fluorescence intensity of EGFP-HSET or Cy5-tubulin within clusters does not increase as a function of time (Fig. 2g). However, this does not rule out the possibility that cluster formation occurs very rapidly on the surface, *i.e.*, on a shorter time scale than the 100 ms exposures used for imaging. We therefore feel it is necessary to refrain from commenting on the precise mechanism of cluster formation in this case.

Because we speculate that cluster formation occurs in solution (rather than on the surface) we reasoned that the suggested TIRF approach is not likely to yield meaningful quantitative results.

Perhaps a more fruitful experimental approach would be FCS performed in solution (*e.g.*, Gaglio, Lahav et al., 2013, *PNAS*), but such an undertaking would be outside the scope of the current manuscript. Importantly, we expect that AMPPNP would lock HSET molecules to the microtubule surface via their motor domains, and thus prevent any molecular rearrangement potentially necessary for cluster formation after the addition of soluble tubulin (in other words, we expect that AMPPNP might actually *prevent* cluster formation). Alternatively, a washout of AMPPNP and replacement with ATP would effectively mimic the TIRF experiments described in the manuscript.

Nevertheless, the suggested experiment indeed yields an important answer to the following question: is it possible for AMPPNP-locked HSET to recruit soluble tubulin to the polymer surface? If successful, a subsequent washout of soluble tubulin and measurements of unbinding kinetics would provide insight into the transience or permanence of the interaction, as indicated by the reviewer. To address this, we first immobilized GMPCPP-seeds labeled with TMR (near-red) via an anti-rhodamine antibody. We then introduced 20 nM EGFP-HSET in the presence of 1 mM AMPPNP to lock saturating amounts of motor to the MT surface. Finally, in the presence of 1 mM AMPPNP, we washed out EGFP-HSET, introduced 1 μ M Cy5-tubulin (70% labeling efficiency), incubated at 35°C for 30 minutes, and imaged the surface by TIRF (see below). As expected, we observed EGFP-HSET bound to the length of the MT lattice. While we did occasionally observe polymeric Cy5-tubulin that was negative for EGFP-HSET (presumably a result of spontaneous nucleation), we did not observe any colocalization between Cy5-tubulin and the seeds:

This result was readily apparent after contrast-adjusting the Cy5 channel to detect weaker interactions:

We therefore conclude that AMPPNP-bound HSET is unable to recruit soluble tubulin to the MT surface, and that cluster formation is likely occurring in solution. Per reviewer suggestion, we now mention this in the discussion and cite the fluorescence intensity result as supporting evidence.

What do the authors think limits the size of the clusters?

Unfortunately, we can merely speculate on this as of now. The most likely possibility is that the clusters form in a closed configuration that cannot elongate indefinitely (a spherical configuration, for instance, should behave like this). Our only evidence for this at the moment is that these clusters appear to be radially symmetric by TIRF imaging. As outlined in the previous rebuttal, we carried out a number of biochemical experiments to address this point, as suggested by the reviewers, but they were inconclusive for reasons we discussed at length in the previous rebuttal letter.

If the behavior described cannot be shown, the only alternative hypothesis I can think of is that tubulin is partly present in an aggregated form (actually still my favorite hypothesis). Even if only a small fraction of the total tubulin is aggregated, these aggregates (after collecting multiple HSET motors) will be fished out of solution by their prolonged interaction with the surface-immobilized microtubules. In a way, that is what the authors somehow show in their model in Figure 7c. Moreover, I believe the presence of such aggregates (or microtubule fragments) is not unlikely, both in vitro and in vivo.

This is a difficult point to address, as absence of proof is not proof of absence. As the reviewer points out, we have “rule[d] out that aggregation of either tubulin or HSET alone is present and [that they are] necessary for the formation of the tubulin-HSET clusters.” We absolutely agree that tubulin clusters may be present at an extremely low level, so much so that they escaped our detection, even with the use of single molecule techniques applied to thousands of tubulin dimers.

We are in agreement that some population of non-canonical tubulin (*i.e.*, not a standard heterodimer or polymeric microtubule but a small oligomer of intermediate size) is likely to be present both *in vitro* and *in vivo*. However, we are hesitant to classify this tubulin as “aggregated,” because SEC-purified tubulin was still able to rescue MT self-organization (see the previous rebuttal), and any aggregates would not co-elute with dimeric tubulin. This leaves three possibilities. Either: (i) a small fraction of the total tubulin *re-aggregated* after SEC purification in a way that was sufficient to activate HSET motility (certainly possible), (ii) a small fraction of the total tubulin is able to convert between canonical and non-canonical forms (*i.e.*, “aggregated”), or (iii) these tubulin aggregates are actually *formed* by HSET. We currently have no way of testing (iii), as the only way we have been able to detect non-canonical tubulin thus far is when it is being transported by HSET in the TIRF assay. In the case of (i) or (ii), these possibilities are extremely interesting and ripe for further study. Perhaps HSET could be a valuable reagent for enriching this population of non-canonical tubulin in subsequent studies.

The speculation that “The lack of a defined cluster structure in our experiments may indicate complex formation through multivalent weak interactions resulting in protein condensation, as previously described in multiple intracellular contexts” is quite far-fetched, as the experimental conditions for observing such condensates *in vitro* are quite different from the present ones.

We agree with the reviewer and have removed this from the discussion.

3) It has been shown before that microtubule destabilizing drugs, such as nocodazole, can induce aster formation (e.g. Visualization of Aberrant Perinuclear Microtubule Aster Organization by Microtubule-Destabilizing Agents, Shinji SAKAUSHI, Kaori SENDA-MURATA, Shigenori OKA & Kenji SUGIMOTO, *Bioscience, Biotechnology, and Biochemistry* 73(5), 1192-1196, 2009). This paper should be cited and discussed.

We thank the reviewer for their suggestion and have now cited this paper. Certainly, aster-promoting factors are still present in our HeLa cells (including endogenous HSET) and we expect that there will always be a baseline level of aster formation as a result of partial microtubule destabilization. Indeed, in our control cells where HSET is not overexpressed, we do observe a slight increase in aster formation after just 15-30 minutes of nocodazole treatment (compared to 6-24 hours in the cited paper). In our case (Fig. 6), we feel confident that HSET overexpression specifically enhances aster formation far above these baseline levels.

It should be noted that the cited paper does not explore the same parameter space as our study, and thus cannot be directly compared. These authors observed aster formation with 100 nM nocodazole after 6-24 hours, and we are using 500 nM nocodazole on the time scale of 15-30 minutes (note that the authors reported to observe aster formation on a ~minutes time scale using 100 nM nocodazole, though this result was not shown).

Further comments:

4) Line 20: “growing MTs” ... I guess it is more important emphasize the presence of free tubulin

We have modified the text accordingly.

5) Lines 106-108: “and that tail deletion is insufficient to form a constitutively active motor.” I believe this statement refers to potential auto-inhibition (as explained later). This may, however, not be clear to the reader at this point.

We have added a clarifying statement in the text.

6) Lines 118-125: I did not get the idea why the authors wanted to visualize single HSET molecules here. Why would that be necessary as compared to imaging the clusters?

Our goal for this experiment was to quantify motor activation by a second method, in addition to imaging clusters. As stated in the manuscript, it was difficult to observe individual events when 100% of the HSET was EGFP-labeled. Because we wanted to quantify single-molecule MSDs of processive HSET (as compared to HSET in the absence of tubulin), the 5% labeling ratio was necessary for this experiment.

7) Line 129: replace “far-red” by “Cy5”

We have modified the text accordingly.

8) Figure 7b: Why would HSET bound to the microtubule via its motor domain diffuse on the microtubule (small arrows)?

Our original goal was to indicate that motor-bound HSET would exhibit less diffusion than tail-bound HSET (hence, the smaller arrows). However, reviewer 2 raises an important point -- we have very little evidence that motor-bound HSET undergoes bidirectional diffusion at all (see black curve, Fig. 2c). We have therefore removed the small arrows from the motor-bound HSET, and we apologize for the oversight.

9) A recent paper in Nature Chemical Biology (doi:10.1038/nchembio.2495) reports on the activity of HSET during microtubule-microtubule sliding (along with a single-molecule characterization of HSET, see Fig. 3). Although there is no direct overlap to the scope of the

current manuscript, the authors may want to consider discussing that paper (for example with regard to the finding that HSET appears to interact slightly differently with microtubules as compared to other kinesin-14s, such as Ncd).

We thank the reviewer for pointing out this beautiful paper that was just published. We agree that there is no overlap to the scope of the current manuscript, but we have now cited their single-molecule characterization where appropriate. For the purposes of framing our current manuscript, we feel that this paper is best grouped together with the other two papers showing that Klp2/Ncd organizes MT bundles to provide a sliding force *in vitro* (Fink et al., 2009, and Braun et al., 2009). As such, we have now cited this new paper alongside the others when making this point. While it is true that HSET may interact slightly differently with microtubules, it remains to be tested whether soluble tubulin drives aster formation in these other species, and we have thus refrained from comment.

Reviewers' comments:

Reviewer #2 (Remarks to the Author):

This re-revised manuscript solved most of my comments from the last round of review and thus presents an improved version. However, with regard to the most important comment (Point 2: What is the nature of the HSET-tubulin clustering?) the additional experiments and the elaborated discussion in the rebuttal letter did - in my eyes - unfortunately and disappointingly not lead to an advance but rather added to the existing concerns. I believe the authors agree with me on that assessment. While I fully understand that answering this question is not easy and that the presented results on their own (addition of a tubulin-containing solution clusters HSET, leading to processive motility and aster formation) are striking, I just do not see that given the present data it can be stated that "soluble" tubulin is doing the job!

The authors state (for example in the abstract): 'HSET binds soluble (non-polymer) tubulin via its N-terminal tail domain to form heterogeneous HSET-tubulin clusters'. Yet, in their paper and in the new experiments, they fail to see (soluble) tubulin binding to the tail domains of HSET. Rather, they do argue, that there might indeed some non-canonical (aggregated) tubulin be present (or: 'tubulin dimers may be interconnected' (without HSET)) and this might be involved in causing the HSET-tubulin clustering. This has been exactly my argument in both rounds of previous review: Can it be ruled out that tubulin aggregates (or small polymers) are involved in the clustering process? If not, all findings can be easily explained but it would need to be shown that such aggregates are relevant in vivo and that the presented in vitro results are not merely an artefact.

Please do not get me wrong, I do not want to be overly picky here. However, I would like to make sure that the authors do not make statements that are not justified and that they might regret later on.

Given the present data, the only possibility I see to describe the data is to openly admit that - despite a number of efforts to avoid their presence - some non-canonical (aggregated) tubulin may be present and responsible for the clustering. If such aggregates were present in cells (and there might be reasons to assume so) they could tune the behavior of HSET. Of course, the whole story is significantly less exciting this way (compared to the original claims) but would still add to our picture of what might be going on in a cellular context.

Further comments:

- Disappointingly, the authors did not come up with any other ideas to further investigate the nature (e.g. permanent vs. transient) of the HSET-tubulin interaction. How about removing the free tubulin out of solution during the processive run of an HSET-tubulin cluster? Wouldn't one expect to see the disintegration of the cluster if the binding was transient?

- I did not mean to say that cluster formation occurs on the microtubule surface, but rather suggested to use microtubules just to immobilize HSET for imaging (and block their motor domains while keeping the tails presented to the solution). And yes, while AMPPNP might indeed prevent cluster formation one should see at least the binding of single tubulins to it. If binding was prevented by too densely loading HSET onto the microtubules (which I hardly believe given the extremely low GFP-signal presented in the rebuttal letter), lower HSET concentration should be used.

- With regard to my question about the maximum size of the clusters, the authors state that the clusters appear radially symmetric in TIRF ... and that this would indicate some spherical structure. Can the authors substantiate this claim given the limited resolution? What is the ellipticity of the imaged particles?

Reviewer #3 (missing from previous decision letter)

The revised manuscript from Ohi and coworkers describes the results of experiments to understand the regulation of HSET motor activity. The data show that a mixture of polymerization competent tubulin and HSET results in aster formation. Both the motor and tail domains are required for aster formation. Single molecule TIRF experiments further show that individual HSET motors are diffusive on stable microtubules and become processive when soluble tubulin is present. Double labeling experiments show that tubulin localizes with the motile EGFP-HSET particles. Data from TIRF experiments is presented to understand the stoichiometry of the motor/tubulin complexes, to show that HSET bound to beads can induce aster formation, and finally to examine aster formation in nocodazole and HSET overexpressing cells.

For the most part the authors have addressed my concerns. In the cellular experiments designed to examine aster formation, the authors overexpress HSET to study the effect of increased number of motors and also use nocodazole to increase the level of soluble tubulin. The results show that more asters are present in nocodazole treated cells overexpressing HSET, but not in nocodazole treated cells treated with DMSO. Both of these conditions have elevated tubulin, but excess asters were observed only when HSET was overexpressed. This experiment, as designed, suggests that elevated tubulin alone does not induce extra asters, but that an excess of motor (perhaps stabilizing the microtubules – there are strong bundles in the Doxy + nocodazole cell, lower panel of C) is also required. Since soluble tubulin is presumably always present *in vivo*, I'm not sure what the physiological relevance is, except to say that as the spindle disassembles in nocodazole, asters and bundles form when there is excess HSET. When excess HSET is not present, the spindle just disassembles rapidly. The authors should consider removing the "significantly" modifier from the description of this experiment in the abstract.

In the final figure the authors show a diagram of what might be occurring during aster formation. I have some concern about the diagram and the text.

First (page 8) the authors state that the 'range' of motors in a cluster is 2-6; based on the average values in the histogram in Figure 3, it seems that the values are 3-4 for HSET and for tubulin 10-12. Consistent with a small number of motors in a cluster, Jonsson et al found that as few as two dimers (plant kinesin 14) show processive motility. Second, if the formation of the motor/tubulin clusters does not depend on tubulin polymerization, as indicated by the new data in which GTP was omitted or colchicine added, why does the diagram show what looks like a short microtubule segment? The tubulin could be in the form of a protofilament or a ring, structures that have been observed previously, not a short microtubule. Indeed, given the challenges in detecting HSET/tubulin complexes in the SEC or AUC experiments, it seems possible that a pair of dimeric motors linked by a tubulin dimer or dimers could be sufficient for switching to processive motion. These options are not compatible with the diagram as drawn. Another concern has to do with the idea that the "activity of HSET is context dependent being influenced directly by the availability of soluble versus polymeric tubulin". Because microtubules are dynamic, they are in equilibrium with soluble tubulin, and that concentration of tubulin should equal the Critical concentration; what is the evidence that in mitotic cells the level of tubulin changes in time or space? The critical concentration could differ for mitosis and interphase, but HSET is nuclear in interphase. Thus it is not clear if tubulin availability is changing. This section should be edited as well as the abstract.

It has been shown that soluble tubulin regulates the severing protein Katanin, and this might be a good reference to add.

In figure 3, the bottom and top panels of figure 6C are incorrectly referred to in the text. In figure 2, there is no panel h. In the reference list, the reference to Jonsson is incomplete.

In figure S3, why not use microtubule bound motors instead of surface bound, which could be

inactive motors.

In the legend to figure 5, the black and red panels are upper and lower.

Reviewer #2 (Remarks to the Author):

This re-revised manuscript solved most of my comments from the last round of review and thus presents an improved version. However, with regard to the most important comment (Point 2: What is the nature of the HSET-tubulin clustering?) the additional experiments and the elaborated discussion in the rebuttal letter did - in my eyes - unfortunately and disappointingly not lead to an advance but rather added to the existing concerns. I believe the authors agree with me on that assessment. While I fully understand that answering this question is not easy and that the presented results on their own (addition of a tubulin-containing solution clusters HSET, leading to processive motility and aster formation) are striking, I just do not see that given the present data it can be stated that “soluble” tubulin is doing the job!

We are very pleased that the 3rd revision of our manuscript further successfully addressed most of Reviewer’s #2 comments, and that the Reviewer finds that ‘*..the presented results on their own ... are striking*’. We think that the main remaining issue is primarily of semantic nature, involving the definition of ‘soluble tubulin’, as elaborated below.

The authors state (for example in the abstract): ‘HSET binds soluble (non-polymer) tubulin via its N-terminal tail domain to form heterogeneous HSET-tubulin clusters’. Yet, in their paper and in the new experiments, they fail to see (soluble) tubulin binding to the tail domains of HSET. Rather, they do argue, that there might indeed some non-canonical (aggregated) tubulin be present (or: ‘tubulin dimers may be interconnected’ (without HSET)) and this might be involved in causing the HSET-tubulin clustering. This has been exactly my argument in both rounds of previous review: Can it be ruled out that tubulin aggregates (or small polymers) are involved in the clustering process? If not, all findings can be easily explained but it would need to be shown that such aggregates are relevant in vivo and that the presented in vitro results are not merely an artefact.

It seems to us that there is a misunderstanding over the definition of “soluble” tubulin. Since our very first manuscript submission, we have defined and used the term “soluble” tubulin as **tubulin present in solution under depolymerizing conditions (i.e., not classic MT polymer)**. Indeed, even in our abstract, as quoted here by the reviewer, we specify “*soluble (non-polymer)*”. In evaluating this point it is important to stress that, over three rounds of extensive reviews, we have performed every experiment suggested by the reviewers to establish that non-polymer tubulin activates clustering and aster formation. Specifically, this included assays in the presence and absence of different GTP-nucleotide analogues; known microtubule depolymerization agents; as well as an additional tubulin purification step involving gel-filtration. Based on all of these experiments, we are confident that non-polymer tubulin is involved in HSET-tubulin cluster formation. However, we acknowledge that Reviewer #2 might equate the term “soluble” with canonical heterodimers of tubulin only. To hopefully reconcile this issue, we have now explicitly added our definition of “soluble” tubulin, which we trust will address Reviewer #2’s main criticism.

Please do not get me wrong, I do not want to be overly picky here. However, I would like to make sure that the authors do not make statements that are not justified and that they might regret later on.

Given the present data, the only possibility I see to describe the data is to openly admit that – despite a number of efforts to avoid their presence - some non-canonical (aggregated) tubulin may be present and responsible for the clustering. If such aggregates were present in cells (and there might be reasons to assume so) they could tune the behavior of HSET. Of course, the whole story is significantly less exciting this way (compared to the original claims) but would still add to our picture of what might be going on in a cellular context.

We **completely agree**, and have at no point actually disagreed with Reviewer #2's assessment that non-canonical clusters of tubulin might be involved in the HSET-tubulin clustering process. We would, however, like to reiterate that we have pursued all reviewer-suggested approaches to characterize both tubulin alone, as well as the HSET-tubulin clusters (including analytical ultracentrifugation, single-molecule fluorescence microscopy, SEC-MALS, and EM), and in particular, we have found no evidence for tubulin aggregates being present in our tubulin-alone solution. Nevertheless, once again, we acknowledge the possibility of very small amounts of tubulin-alone clusters not detectable by these methods could be present.

Where we **completely disagree** with Reviewer #2, is on their assessment of the significance of our findings. While the precise nature of the assembly and structure of HSET-tubulin clusters has not been resolved and will clearly require significant time investment and technologies more sensitive than what we have used here, we strongly believe that the concept of switching motor activity, and consequently microtubule network morphology, through relative availability of non-polymeric tubulin is absolutely novel, and highly significant. Given the positive assessments from the other two expert reviewers, as well as Reviewer #2's own statement that *'the presented results on their own ... are striking'*, we feel that Reviewer #2's assessment of the significance is, at best, subjective.

In this regard, we would like to particularly underscore the exciting results we have obtained from cellular experiments during the revision process. Our results in cells provide important insight into the physiological relevance of our findings. In our opinion, Reviewer #2's assessment alone that our results *'still add to our picture of what might be going on in a cellular context'*, strongly advocates for their significance.

Further comments:

- Disappointingly, the authors did not come up with any other ideas to further investigate the nature (e.g. permanent vs. transient) of the HSET-tubulin interaction. How about removing the free tubulin out of solution during the processive run of an HSET-tubulin cluster? Wouldn't one expect to see the disintegration of the cluster if the binding was transient?

Once again, we would like to stress out that we have completed several rounds of extensive revisions, and performed every experiment that was suggested by reviewers. Given this, while we are excited to continue pursuing further ideas, we feel that the additional detailed studies of the exact nature of the HSET-tubulin clusters are beyond the current scope of our manuscript.

- I did not mean to say that cluster formation occurs on the microtubule surface, but rather suggested to use microtubules just to immobilize HSET for imaging (and block their motor domains while keeping the tails presented to the solution). And yes, while AMPPNP might indeed prevent cluster formation one should see at least the binding of single tubulins to it. If binding was prevented by too densely loading HSET onto the microtubules (which I hardly believe given the extremely low GFP-signal presented in the rebuttal letter), lower HSET concentration should be used.

If this strategy was not proposed to visualize the assembly of clusters, then it is not superior, in our opinion, to the approach we already used to examine the composition of HSET-tubulin clusters (Figure 3). In Figure 3, we determined the fluorescence intensities of tubulin and HSET in clusters following their engagement with the microtubule lattice.

As we explained in the last revision, there is strong reasoning behind the possibility that Reviewer #2's proposed experiment would not work. Reviewer #2 likely assumes that, if the HSET motor domain was bound to polymer, the exposed tail domain should bind soluble tubulin with affinity that is equal to (or not greatly diminished from) its ability to bind soluble tubulin in solution. However, we do not know if the tail domain changes conformation after the motor domain binds polymeric MTs. Additionally, we (and others) have shown that the tail domain also independently binds polymeric MTs. Thus, it is possible that HSET could adopt a "lying down" conformation on the polymer surface, minimizing the chance that the tail could bind tubulin from solution. As a final point, we have never proposed that the HSET tail necessarily binds an individual heterodimer, as we have outlined in the previous paragraph.

- With regard to my question about the maximum size of the clusters, the authors state that the clusters appear radially symmetric in TIRF ... and that this would indicate some spherical structure. Can the authors substantiate this claim given the limited resolution? What is the ellipticity of the imaged particles?

We thank the reviewer for this point. In our manuscript, we made no claims about a spherical structure. Reviewer #2 appears to be addressing this section of our previous rebuttal:

"Unfortunately, we can merely speculate on this as of now. The most likely possibility is that the clusters form in a closed configuration that cannot elongate indefinitely (a spherical configuration, for instance, should behave like this)."

Our sole intention that the clusters appear radially symmetric in TIRF is to point out that the clusters are diffraction-limited, *i.e.*, small. We meant to use a spherical configuration as an

example and regret if this was interpreted in a different way. Indeed, we have no reason to believe that this structure is spherical, cylindrical, or otherwise. Given reviewer #3's point about our model figure (how can we assume the clusters are a given shape?), we agree that we need to be more clear in the manuscript about this. As such, we have included a clarifying sentence in the discussion and in the legend for the model figure. We thank the reviewers for pointing this out.

Reviewer #3 (missing from previous decision letter)

The revised manuscript from Ohi and coworkers describes the results of experiments to understand the regulation of HSET motor activity. The data show that a mixture of polymerization competent tubulin and HSET results in aster formation. Both the motor and tail domains are required for aster formation. Single molecule TIRF experiments further show that individual HSET motors are diffusive on stable microtubules and become processive when soluble tubulin is present. Double labeling experiments show that tubulin localizes with the motile EGFP-HSET particles. Data from TIRF experiments is presented to understand the stoichiometry of the motor/tubulin complexes, to show that HSET bound to beads can induce aster formation, and finally to examine aster formation in nocodazole and HSET overexpressing cells.

For the most part the authors have addressed my concerns. In the cellular experiments designed to examine aster formation, the authors overexpress HSET to study the effect of increased number of motors and also use nocodazole to increase the level of soluble tubulin. The results show that more asters are present in nocodazole treated cells overexpressing HSET, but not in nocodazole treated cells treated with DMSO. Both of these conditions have elevated tubulin, but excess asters were observed only when HSET was overexpressed. This experiment, as designed, suggests that elevated tubulin alone does not induce extra asters, but that an excess of motor (perhaps stabilizing the microtubules – there are strong bundles in the Doxy + nocodazole cell, lower panel of C) is also required. Since soluble tubulin is presumably always present in vivo, I'm not sure what the physiological relevance is, except to say that as the spindle disassembles in nocodazole, asters and bundles form when there is excess HSET. When excess HSET is not present, the spindle just disassembles rapidly. The authors should consider removing the “significantly” modifier from the description of this experiment in the abstract.

We thank the reviewer for his/her feedback and have implemented the suggested text edit. Indeed, an excess of both motor and tubulin are required for multiple asters to form simultaneously. Our goal for this experiment was not to establish multi-aster formation as a physiological event, but rather, that HSET and soluble tubulin *can override endogenous* spindle assembly/disassembly factors in cells, under the right circumstances (*ie.*, high concentrations of both motor and tubulin). We therefore interpret this as compelling evidence that HSET may use our proposed mechanism to promote MT-minus end pole focusing in the context of the cellular milieu, particularly as the levels of non-polymeric to polymeric tubulin might be modulated (*eg.*, in the case of cancer cells with supernumerary centrosomes).

Additionally (and very relevant to Reviewer #2's concerns), the cell-based experiments provide clear proof of principle that our proposed mechanism does not simply rely on tubulin aggregates that are an artifact of our tubulin preps.

In the final figure the authors show a diagram of what might be occurring during aster formation. I have some concern about the diagram and the text.

First (page 8) the authors state that the 'range' of motors in a cluster is 2-6; based on the average values in the histogram in Figure 3, it seems that the values are 3-4 for HSET and for tubulin 10-12. Consistent with a small number of motors in a cluster, Jonsson et al found that as few as two dimers (plant kinesin 14) show processive motility.

Given the fluorescence intensity distributions in Figure 3, an estimate of 3-4 for HSET and 10-12 for tubulin is far too narrow a range for these complexes. We have modified the text to "possible range" rather than "typical range," to eliminate confusion. The point about two dimers is well-taken, and we have consistently argued that as few as two motors is sufficient for activation.

Second, if the formation of the motor/tubulin clusters does not depend on tubulin polymerization, as indicated by the new data in which GTP was omitted or colchicine added, why does the diagram show what looks like a short microtubule segment? The tubulin could be in the form of a protofilament or a ring, structures that have been observed previously, not a short microtubule. Indeed, given the challenges in detecting HSET/tubulin complexes in the SEC or AUC experiments, it seems possible that a pair of dimeric motors linked by a tubulin dimer or dimers could be sufficient for switching to processive motion. These options are not compatible with the diagram as drawn.

We agree with the reviewer's point that we do not know what the morphology of the HSET-tubulin clusters is, and we explain this in the text and figure legend. Although we are completely agnostic when it comes to the cluster structure, we do feel that having a model figure for illustration purposes is valuable, and we have modified the model figure since the original submission for conceptual clarification as suggested by previous reviews. Furthermore, while the idea that a pair of dimeric motors could be linked by a single tubulin dimer is not impossible, this is extremely unlikely. We never observed single tubulin dimers transported by HSET, and tubulin is present in stoichiometric excess to HSET in every circumstance we tested.

Another concern has to do with the idea that the "activity of HSET is context dependent being influenced directly by the availability of soluble versus polymeric tubulin". Because microtubules are dynamic, they are in equilibrium with soluble tubulin, and that concentration of tubulin should equal the Critical concentration; what is the evidence that in mitotic cells the level of tubulin changes in time or space? The critical concentration could differ for mitosis and interphase, but HSET is nuclear in interphase. Thus it is not clear if tubulin availability is changing. This section should be edited as well as the abstract.

This a very important point, and we are happy to clarify this. In isolation, we agree that microtubules are in equilibrium with soluble tubulin. However, this relationship is much more complex in the context of the cell. We already know that many microtubule-associated proteins (MAPs) associate with soluble tubulin (*eg.*, Op18), or have the capacity to bind both soluble tubulin and the microtubule lattice (*eg.*, kinesin-8s, and TOG-family proteins). Thus, following microtubule disassembly, soluble tubulin has the potential to engage these protein factors, in addition to becoming reincorporated into the microtubule lattice. In other words, tubulin availability has the potential to change EVERY time a microtubule disassembles. In this scheme, reentry of unpolymerized tubulin into a polymerization-competent pool thus depends on the off rate of tubulin and MAPs. Whether tubulin engagement with MAPs is regulated throughout the cell cycle is also an interesting issue. As the reviewer correctly points out, HSET-tubulin interactions are antagonized during interphase *via* nuclear sequestration of HSET. Additionally, cancer cells with supernumerary might have altered levels and balance of non-polymerized to polymerized tubulin.

It has been shown that soluble tubulin regulates the severing protein Katanin, and this might be a good reference to add.

We agree with the reviewer and now reference this paper in the discussion.

In figure 3, the bottom and top panels of figure 6C are incorrectly referred to in the text. In figure 2, there is no panel h. In the reference list, the reference to Jonsson is incomplete.

We thank the reviewer for pointing these out. We have corrected the errors referring to Figure 2H and the Jonsson citation. In our review of the manuscript, the referencing to Figure 6C is correct. We are unsure of what Reviewer #3 is referring to.

In figure S3, why not use microtubule bound motors instead of surface bound, which could be inactive motors.

We have used surface-bound molecules because there is a negligible advantage to using MT-bound motors (inactive motors may still be able to bind MTs, after all). Please note that this is considered a field-standard technique (see, for instance, the Jonsson paper that was referenced a few sentences ago).

In the legend to figure 5, the black and red panels are upper and lower.

We thank the reviewer for his/her attention to detail and have fixed this text.